# A sensitivity study on the measurement of urban polycentricity in chinese cities: Center definition, indicator selection, and their interaction effects

Juan Zhu[1]*, Yao Wang[2]

**1** College of Art and Design, Suzhou Chien-shiung Institute of Technology, Suzhou, Jiangsu Province, China, **2** College of Architecture and Urban Planning, Suzhou University of Science and Technology, Suzhou, Jiangsu Province, China

\* zhujuan@csit.edu.cn

## Abstract

Research on urban polycentricity has long been hampered by fragmented case studies, inconsistent standardized criteria, and methodological misuse, complicating cross-study comparisons and limiting a nuanced understanding of complex spatial structures. To systematically reveal the combined effects of center definition methods and measurement indicators on polycentricity assessment, this study takes Hangzhou, Wuhan, and Nanning as comparative cases. We integrate four employment center identification methods and five polycentricity measurement indicators to conduct a comprehensive assessment from both morphological and functional dimensions. A Linear Mixed-effects Model (LMM) is employed to quantify the influence of method choices, indicator properties, and their interactions on the measurement results. The findings indicate that: (1) There are systematic differences between morphological and functional polycentricity. Morphological polycentricity is insensitive to the choice of center definition methods, whereas functional polycentricity is significantly constrained by the identification method used. (2) A significant "method-indicator" interaction effect exists in functional polycentricity. The LMM reveals that specific "method-indicator" combinations systematically lead to substantial variations in measurement results, suggesting that functional polycentricity is a "dynamic process" co-constructed by methodological choices rather than a purely objective reality. (3) Inter-city comparisons reveal diverse pathways of polycentric development: Hangzhou exhibits a "mature-type" structure characterized by the synergy between morphological and functional polycentricity; Wuhan demonstrates a transitional pattern of "morphologically monocentric but functionally networked"; while Nanning represents a policy-driven emerging polycentric city. This study argues that methodological choices profoundly shape the conclusions of polycentricity research. Future studies and practices should emphasize methodological transparency and sensitivity analysis, promoting a paradigm shift from "seeking the single truth" to "understanding

**Data availability statement:** The dataset underlying this study consists of proprietary cellular signaling data provided by China Mobile Communications Group Co., Ltd. This dataset contains detailed, aggregated records of mobile device locations and movements. Public sharing of this dataset is restricted due to a legally-binding Data Licensing and Non-Disclosure Agreement between the data provider and our research institution. The agreement explicitly prohibits the redistribution or public deposition of the raw or processed signaling data to protect commercial confidentiality and user privacy. Researchers interested in the underlying data may contact China Mobile directly (https://www.10086.cn).

**Funding:** This research study was supported by General Project of Philosophy and Social Science Research in Colleges and Universities of Jiangsu Province (2023SJYB1622). The funders had no role in study design, data collection and analysis, decision to publish, or preparation of the manuscript.

**Competing interests:** This manuscript has not been published or presented elsewhere in part or in entirety and is not under consideration by another journal. We have read and understood your journal's policies, and we believe that neither the manuscript nor the study violates any of these. There are no conflicts of interest to declare.

sources of uncertainty," thereby providing more reliable foundations for scientific urban planning.

## 1. Introduction

Since the second half of the 20th century, the importance of the traditional Central Business District (CBD) as an employment center has been declining [1]. Particularly since the 1990s, the development of the post-industrial economy has led to the dispersal of employment populations across urban regions, evolving towards a decentralized or polycentric structure [2]. Polycentricity has become a common phenomenon in large cities in the post-industrial era. Concurrently, many municipal governments have adopted polycentric policies to address a series of issues faced by monocentric urban development, such as spatial mismatch between jobs and housing, traffic congestion, and environmental pollution. Polycentricity, both as a spatial phenomenon and as a policy tool, has sparked extensive debate and discussion in academic circles. It remains unclear whether there is an inherent relationship between the two, and it is difficult to find empirical evidence to fully support the positive claims in polycentric policies [3].

Although polycentricity has become a widely accepted concept, its theoretical research framework remains underdeveloped [4]. From an empirical research perspective, determining whether a metropolitan area is polycentric involves four steps: first, delineating the study area; second, determining the unit of analysis; third, identifying centers; and finally, selecting a method to measure polycentricity. Since each of these steps can be approached in multiple ways across different urban studies, polycentricity research inherently carries uncertainty. Firstly, selecting appropriate study boundaries is crucial for the consistency of polycentricity measurement [5,6]. Polycentricity is a scale-sensitive concept, as the spatial scale increases, the level of polycentricity often decreases [6–8].Secondly, the choice of the research unit also influences the measurement of polycentricity. For instance, as the study units of the Tokyo metropolitan area become progressively smaller, the polycentric structure becomes more pronounced [9]. Thirdly, the identification of centers is crucial for measuring polycentricity, significantly affecting the accuracy of the analysis, spatial distribution patterns, policy implications, and comparative studies [10]. Different identification methods can yield different results for the same city. Taking Los Angeles data from 1990 as an example: Giuliano et al. [11], using the Giuliano and Small [12] threshold method, identified 46 centers; Forstall and Green [13], using an jobs-housing ratio method, identified 120 centers; while Redfearn [14] and Lee [15], based on the Local Weighted Regression (LWR) method, identified 41 and 44 centers respectively. Fourth, the measurement of polycentricity is highly sensitive to methodological choices [16–18], different measurement approaches do not necessarily lead to consistent or similar conclusions [18,19]. Furthermore, a city that is polycentric in form is not necessarily polycentric in function. An urban spatial structure should ideally be polycentric in both morphology and function [17].Thus, due to differences in research procedures and standards, the measurement of polycentricity depends on

numerous complex factors. Comprehensively understanding all influencing factors in measuring polycentricity is currently an unattainable, daunting task [10].

From a methodological perspective, the measurement of polycentricity relies on a clear conceptual and methodological foundation. The measurement process requires defining an appropriate scale and method for center identification, and exploring whether different measurement methods can yield consistent or similar conclusions. Due to limitations in the availability of static and dynamic data, traditional studies have often relied on single-dimensional or single-framework approaches focusing solely on either morphology or function. Based on mobile phone signaling data, this study captures both the static distribution of the working population and the dynamic patterns of commuting flows. It aims to explore, from both morphological and functional dimensions, the relationships between different types of center definition methods and measurement approaches, as well as their effects on the assessment of urban polycentricity—an aspect that has been largely overlooked in previous research. To this end, this study aims to achieve the following objectives:

Defining Centers: We aim to investigate the impact of different employment center identification methods on determining urban polycentricity.

Measuring Polycentricity: We aim to analyze the influence of different polycentricity measurement methods on determining urban polycentricity from both morphological and functional dimensions.

Interaction of Factors: We aim to study how center definition, measurement methods, and different case study cities interact to influence the assessment of polycentricity.

To achieve this research objective, we selected three Chinese cities of different scales—Wuhan, Hangzhou, and Nanning—as case studies [20], and proposed a polycentricity measurement framework based on both morphological and functional dimensions（see Fig 1）. The implementation process includes: (1) In the analysis of urban spatial structure, four center identification methods are employed for cross-validation to examine how the spatial locations and boundaries of identified employment centers influence the research outcomes; (2) The morphological and functional dimensions are integrated into a unified identification and measurement process, rather than being considered separately, in order to explore whether spatially dispersed urban forms necessarily lead to functional coordination and balance; (3) For

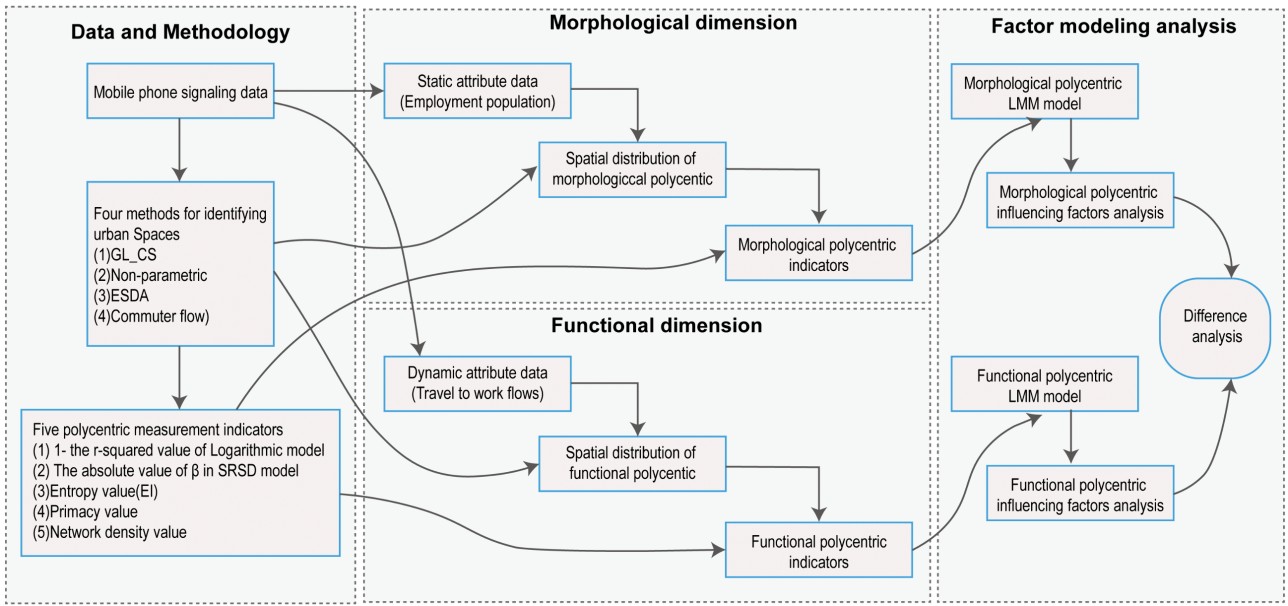

**Fig 1. Technical framework for polycentricity measurement based on morphological and functional dimensions.**

polycentricity measurement, multiple indicators are used to compare and analyze the differences between morphological and functional polycentricity results. Furthermore, a linear mixed-effects model (LMM) is applied, treating polycentricity as the dependent variable, to analyze how center identification methods and measurement indicators affect the measurement results. Compared with previous studies, this research systematically evaluates urban morphological and functional polycentricity using multiple approaches, thereby enriching the understanding of complexity and uncertainty in studies of intra-urban spatial structure and polycentricity.

## 2. Relevant literature review

### 2.1. The ambiguity of the concept of polycentricity

The concept of polycentricity is inherently ambiguous, which is mainly reflected in two aspects: (1) scale issues, and (2) the distinction between function and morphology [17,21].

Regarding the issue of scale, the concept of polycentricity is scale-dependent [16,21,22]. Davoudi [16] identified three levels of polycentricity: intra-urban, inter-urban, and inter-regional. Intra-urban polycentricity, or the polycentric city (PC), refers to multiple centers within a single urban area—for instance, Los Angeles, Paris, London, and Shanghai. Inter-urban polycentricity, or the polycentric urban region (PUR) [23], refers to networks of cities such as the U.S. West Coast, Central Scotland, the Randstad in the Netherlands, the Ruhr area in Germany, and China's Yangtze River Delta. Inter-regional polycentricity refers to national or international spatial structures, for instance, in the European Union's regards polycentricity as a key tool for enhancing regional cohesion and competitiveness [16].Van Meeteren et al. [21] further argue that the conceptual distinction between intra-urban and inter-urban polycentricity is not primarily based on scale, but rather on social context, methodological approach, and applicability. Intra-urban polycentricity focuses on population distribution within equal or larger urban areas, while inter-urban polycentricity concerns multi-city regions integrated through administrative and functional linkages. These correspond to different urbanization models in North America and Europe, respectively. Therefore, in urbanized regions of similar size and population, the boundary between the two concepts is not clearly defined.

The concept of polycentricity encompasses two dimensions: morphological and functional [17,23]. There exists an interdependent relationship between morphological polycentricity (balanced size distribution or absolute importance of centers) and functional polycentricity (balanced distribution or relative importance of functional linkages among centers) [17].In a morphologically polycentric system, there is no dominant center in terms of size or scale; in a functionally polycentric system, no center dominates the network in terms of functional flows—that is, there is no clear directional hierarchy toward a particular center. Morphological polycentricity is commonly assessed using population or employment size data [2,14], and when such data are unavailable, physical spatial indicators such as built environment characteristics are often used instead [24,25]. Functional polycentricity, on the other hand, is typically measured through flow data—such as commuting, leisure and entertainment activities, or financial transactions [26–28].A city that is morphologically polycentric is not necessarily functionally polycentric [17,23]. For example, in the Netherlands, functional polycentricity emerged earlier than morphological polycentricity [17]. Truly polycentric cities should exhibit a high degree of both morphological and functional polycentricity. Empirical research in Western Europe has revealed a relatively strong and positive relationship between these two dimensions [17,29]. However, studies in China have not shown the same pattern [30].

### 2.2. The challenges in measuring polycentricity

The measurement of polycentricity has long lacked standardized methodologies, which directly undermines the effectiveness of academic dialogue and the comparability of research findings. For instance, Liu et al. [30] categorized polycentricity measurement approaches into three major types: social network theory, rank–size distribution, and exploratory spatial analysis methods. Studies have shown that different measurement approaches may yield vastly different conclusions

even for the same urban area. This discrepancy mainly arises from substantial variations in their theoretical foundations, data requirements, and definitions of centers [18].

Morphological polycentricity assesses the evenness of the distribution of center sizes or absolute importance through the attributes or internal characteristics of centers. Its measurement methods primarily include the primacy index [31], rank–size distribution (SRSD) [17,22], and various morphological polycentricity indices. The primacy index assesses the degree of polycentricity by calculating the ratio between the size of the largest center and that of subsequent centers [29]. The rank–size distribution evaluates polycentricity through the slope of the regression line, a flatter slope indicates a more balanced distribution and, thus, a higher degree of polycentricity [6,17,18]. The morphological polycentricity index encompasses a range of indicators such as the entropy index [19,32], the standard deviation ellipse index of population density with four threshold groups [33], a PCA-based index derived from seven employment subcenter population density indicators [34], and network density measures [18,30].

Functional polycentricity measurement assesses the relative importance of centers through the network characteristics formed by their specialized complementarity and interactions. [35,36]. The measurement approaches primarily include flow distribution balance, network connectivity, and commuting typologies. Flow distribution balance methods focus on the evenness of interactions between centers, represented by measures such as the primacy index [18,29], entropy index [19,32], rank–size distribution of internal centrality scores [17] and commuting flow hinterlands [8].Network connectivity evaluates centrality using network-based indicators such as node centrality [36], network flow density [18,23,30], network centrality measures [27], and kernel density [37]. For instance, Green [23] proposed a general functional polycentricity index that combines network density and the distribution of centrality, while Liu et al. [30] later refined this method. However, the approach remains highly sensitive to the number of centers considered—meaning that the selection of centers (cities) used to calculate network density can significantly affect the measurement outcomes. Commuting typology methods include models such as urban commuting system model [38] and urban network structure prototypes, which classify cities based on commuting flow patterns [32,37]. Although morphological and functional measurements are distinct, they are methodologically and empirically interconnected. Methodologically, analytical frameworks such as primacy index, rank–size distribution, entropy index, and network density can be applied to both morphological and functional dimensions, providing a comparative foundation for the two approaches [17–19]. For example, Burger and Meijers [17] used the rank–size method and found that a city region that is morphologically polycentric is not necessarily functionally polycentric; however, functionally polycentric city regions often exhibit certain morphological polycentric characteristics.

With the rise of big data technology, morphological polycentricity assessment has shifted from relying on static indicators such as employment/population density and primacy to increasingly integrating multi-source spatial data such as nighttime light remote sensing [39], high-resolution land use [40], and points of interest (POI) [37], and applying methods such as spatial clustering, spatial econometrics, and morphological analysis. This shift reveals the dynamic evolution of urban physical form, three-dimensional features (e.g., building density, urban skyline), and the intrinsic relationship with the distribution of functional spaces, making the definition of "morphology" more refined and multi-dimensional. Functional polycentricity research has evolved from early analyses of single-direction flows such as commuting and traffic flows to constructing complex urban functional networks based on multi-source dynamic big data, including mobile phone data [28,41,42], ride-hailing/taxi trajectory data [27,37,43], subway smart card data [44,45], corporate linkage data [46], and social media data [26]. Research methods have also shifted from traditional gravity models and flow analysis to complex network analysis [26,46], community detection algorithms [37], and multilayer network models [43,44], aiming to capture multi-level, multi-modal, high-frequency dynamic functional linkages and spatiotemporal evolution within cities. The paradigm of polycentricity research has transitioned from traditional indicator-based measurement to the deep integration of multi-source heterogeneous big data and the cross-application of advanced analytical technologies. For instance, Liu et al. [26] employed an improved Fast-Newman algorithm to construct a multi-dimensional urban functional network using location data from 40 million users in Tokyo; Duan et al. [46] integrated enterprise big data with transportation big data to

develop a dual measurement framework of "network-cluster" in Shenzhen; Yue et al. [43]validated a four-layer functional network structure in Changsha using trajectory data from 1.3 million taxis; Zhang et al. [44] successfully captured the decentralization trend in London from 2013 to 2017 using smart card data. The significant differences observed in studies across different cities largely stem from disparities in data accessibility and the application of analytical techniques. For example, Yue et al. [42] through cross-validation of mobile phone communication data and land development data in Shanghai, found that the number of identified functional centers was significantly lower than that of morphological centers, indicating a lag in planning interventions behind market mechanisms. Similarly, Yu et al. [37] using multi-source data including building data, road networks, POI, and ride-hailing data in Shenyang, combined with GIS spatial analysis and network modeling methods, discovered that morphological centers are concentrated in peripheral areas, while functional centers exhibit bidirectional penetration. Future research on urban polycentricity will be a data-intensive, model-driven, and interdisciplinary field. The core challenge lies in progressing from data visualization to understanding the mechanisms of polycentricity, ultimately achieving the "optimization" of polycentricity through smart planning and governance. In line with this trend, the present study systematically compares the effects of various "method-indicator" combinations, aiming to provide a benchmark reference for methodological standardization and the reliability of conclusions in this field.

## 3. Research areas, data and methods

### 3.1. Research areas and units

This study selected three Chinese cities with varying population sizes and development levels—Wuhan, Hangzhou, and Nanning—as case studies to examine the potential impact of different urban development stages on the measurement of polycentric structures. In 2018, the permanent resident populations of their municipal districts were 11.081 million, 8.57 million, and 4.4176 million, respectively, with per capita GDPs of 187,263 yuan, 135,877 yuan, and 57,781 yuan (National Bureau of Statistics, 2021). Spatially, Wuhan is a central city in central China with a municipal area of 8,494 km²; Hangzhou is a core city in the Yangtze River Delta region, with a municipal area of 16,850 km²; and Nanning is an important central city in southern China with a municipal area of 22,099 km².It is evident that the municipal area of each city varies considerably. Given that the study scale is a crucial factor affecting polycentricity measurements, to avoid the interference of spatial heterogeneity caused by administrative boundaries, this study adopts the concept of Functional Urban Regions (FURs) proposed by Parr [47] and delineates comparable spatial study areas through multiple criteria. The specific criteria include: (1) a commuting flow of at least 5% of the permanent labor force to the city's core area; (2) a population density higher than the city's average and a total population exceeding 5,000; and (3)a clustered road network density area larger than 5 km². Based on these criteria, the final study areas for Hangzhou, Wuhan, and Nanning are 4,091.88 km², 5,026.96 km², and 4,827.73 km², accounting for 24.28%, 59.18%, and 21.85% of their respective municipal areas. This ensures that subsequent analysis are conducted on comparable spatial scales.

The choice of the study unit also profoundly influences the measurement of polycentricity. Generally, the smaller the study unit, the stronger the polycentricity observed [9]. In China, constrained by data availability, most research relies on the street-level units. However, street-level units are relatively large, and the centers identified are often not actual physical employment centers in the physical sense. In contrast, grid units, while capable of identifying finer employment centers [9], may artificially segment continuous urban road network and functional linkages due to their geometric boundaries, leading to discrepancies between identified centers and real built environment boundaries. To address this, this study adopts Traffic Analysis Zones (TAZs) as the basic spatial units. TAZs are typically more aligned with road networks and built-up area forms, better reflecting the natural boundaries of urban internal functional organization. Combined with extensive mobile signaling data, TAZ units enable the identification of smaller-scale employment centers in central urban areas while effectively capturing larger employment centers such as suburban university towns and industrial parks. This allows for the refined identification of polycentric structures at a micro-scale [48], as illustrated in Fig 2.

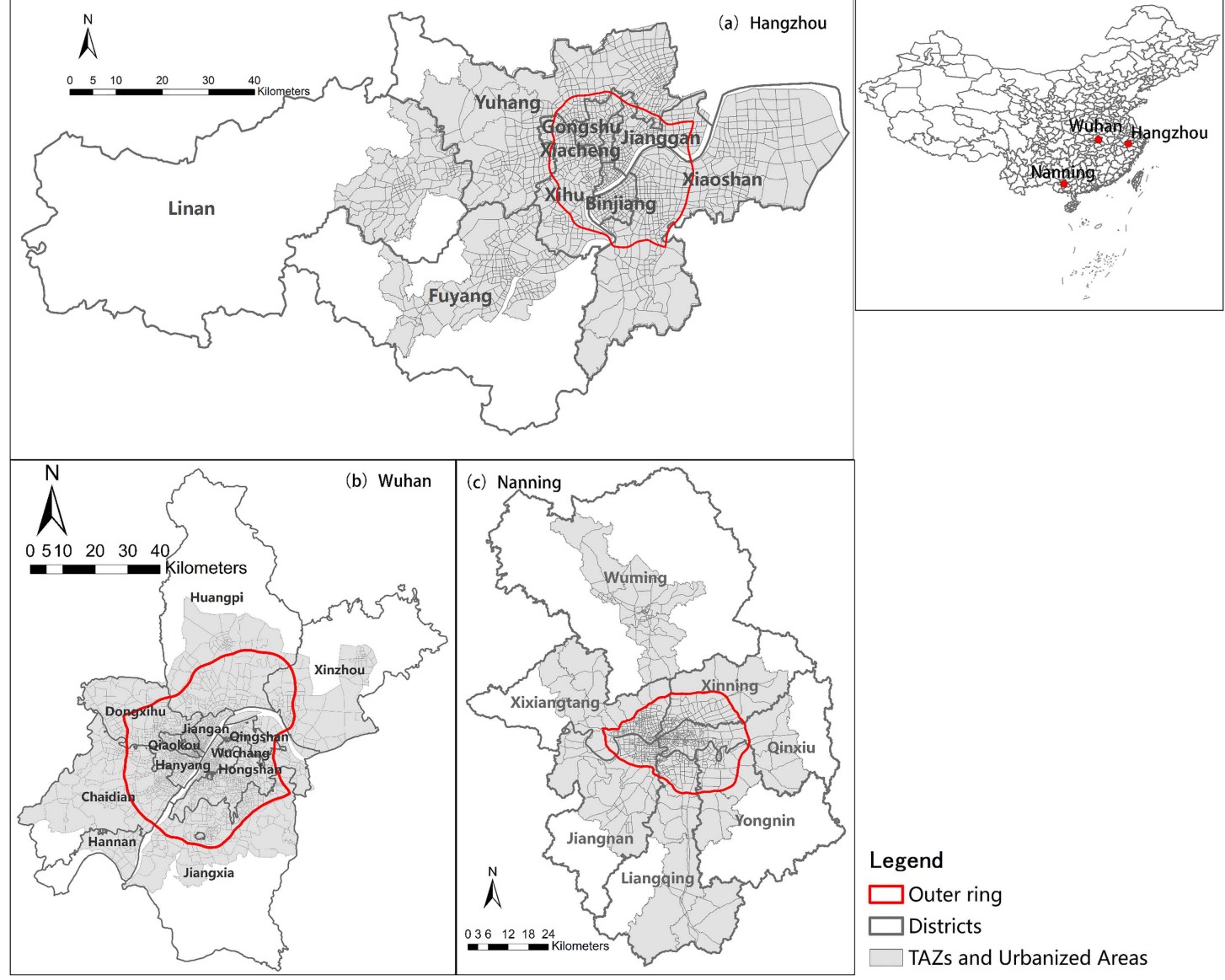

**Fig 2. Research areas and units.** Basemap data sourced from Tianditu (https://map.tianditu.gov.cn/, Ministry of Natural Resources, Map Review No. GS(2025)1508).Administrative and TAZ boundaries are sourced from the respective municipal governments of Hangzhou, Wuhan, and Nanning, and are used with permission.

## 3.2. Research data

This study utilizes one month of continuous mobile phone signaling data from China Mobile for Nanning, Hangzhou and Wuhan, collected in March, April, and August 2017 respectively. The use of data from the same year ensures temporal consistency for cross-city comparisons. Spatiotemporal mobile positioning data can capture approximately 50% of commuting behavior, making it an ideal data source for identifying locations such as workplaces, homes, and schools [49]. This study employs the cumulative time method to identify users' homes and workplaces [49,50]. The identified distribution of working and residential populations provides not only locational information on population distribution but also data on

work-commuting flows, making it suitable as a data source for both morphological and functional polycentricity analysis. First, users' stay points are identified from their activity trajectories recorded in the signaling data. Nearby trajectory points (within 1,000 meters) or temporally close records are merged to construct a stay-point dataset. Second, based on the stay-point dataset, the cumulative time method is used to determine each user's home and workplace: the location where a user stays the longest (for more than one hour) during nighttime (20:00–6:00) and daytime (9:00–16:00), respectively, is defined as their home and work location for that day. Third, users who have repeated identification consistency rates of at least 50% over 30 consecutive days (or 20 weekdays) at the same locations, have non-zero commuting distances (to exclude retirees, home-based individuals, and very short-distance commuters), and for whom both home and workplace are identified, are classified as the commuting population. Through the above processing, 2.2283 million, 1.1986 million, and 824.9 thousand commuting individuals were identified for Hangzhou, Wuhan, and Nanning, respectively. Within these, the proportions of commuting populations falling within the previously defined study areas are 88.47%, 98.57%, and 69.28%, respectively, indicating that the delineated study areas cover regions of highly concentrated population activity within the cities, meeting the scale requirements for studying intra-urban spatial structures.

### 3.3. Research methods

**3.3.1. Identifying employment centers.** The location and boundaries of employment centers are a prerequisite for analyzing the spatial structure of cities [24]. However, employment centers often lack fixed and clearly defined boundaries. Existing studies usually identify centers based on the spatially static characteristics of the physical environment and resource distribution, such as the density threshold method [12,51], non-parametric methods [52], and spatial statistical approaches [53]. Some studies have also used flow-based spatial approaches to identify centers, such as the commuting flow method and community detection algorithms [27,54,55].To examine how different methods of identifying employment centers affect the measurement of urban polycentricity, this study applies four approaches: the threshold method (GL_CS), the non-parametric method, the spatial statistical method (ESDA), and the commuting flow method. Detailed methodological descriptions and implementations can be found in the work of Zhu et al. [20]. The specific results of center identification and commuting flows are shown in Fig 3–5. By comparing four different employment center identification methods across three cities at varying stages of economic development, this study explores whether differences in the number and spatial distribution of identified centers influence the measurement of polycentricity, thereby enhancing the scientific robustness of the empirical analysis.

**3.3.2. Measuring polycentricity.** This study employs five well-established indicators with solid theoretical foundations and empirical validation in the polycentricity literature to comprehensively measure urban polycentricity from both morphological and functional dimensions (definitions provided in Table 1). These indicators include: the Primacy Index (PM) [29], the Slope of the Rank–Size Distribution (SRSD) [17,22,29,30], Network Density [23,30], the Entropy Index (EI) [32,56], and the logarithmic model R² [8]. These indicators have been widely applied to compare the morphological and functional polycentricity of urban systems across different regions, particularly in Northwestern Europe and China [18,19,30,42].

Although the measurement methods for morphological and functional polycentricity are similar, the underlying data used for each differ considerably. Morphological polycentricity is usually based on a single type of data, typically employment or population density. In contrast, functional polycentricity involves a more diverse set of data sources. For example, the Primacy Index [18,29] and the Rank–Size Distribution Slope [6,17] are based on inflow commuting volumes to centers. The Network Density [18,19,32] and Entropy Index [18,19,32] use total commuting volumes of centers, while the logarithmic model's R² value employs the employment center linkage strength indicator [8], which uses the R² statistic from Ordinary Least Squares (OLS) regression to estimate the relationship between the central connection field and the potential field. An R² value closer to 1 indicates a stronger functional connection between that center and other parts of the urban system.

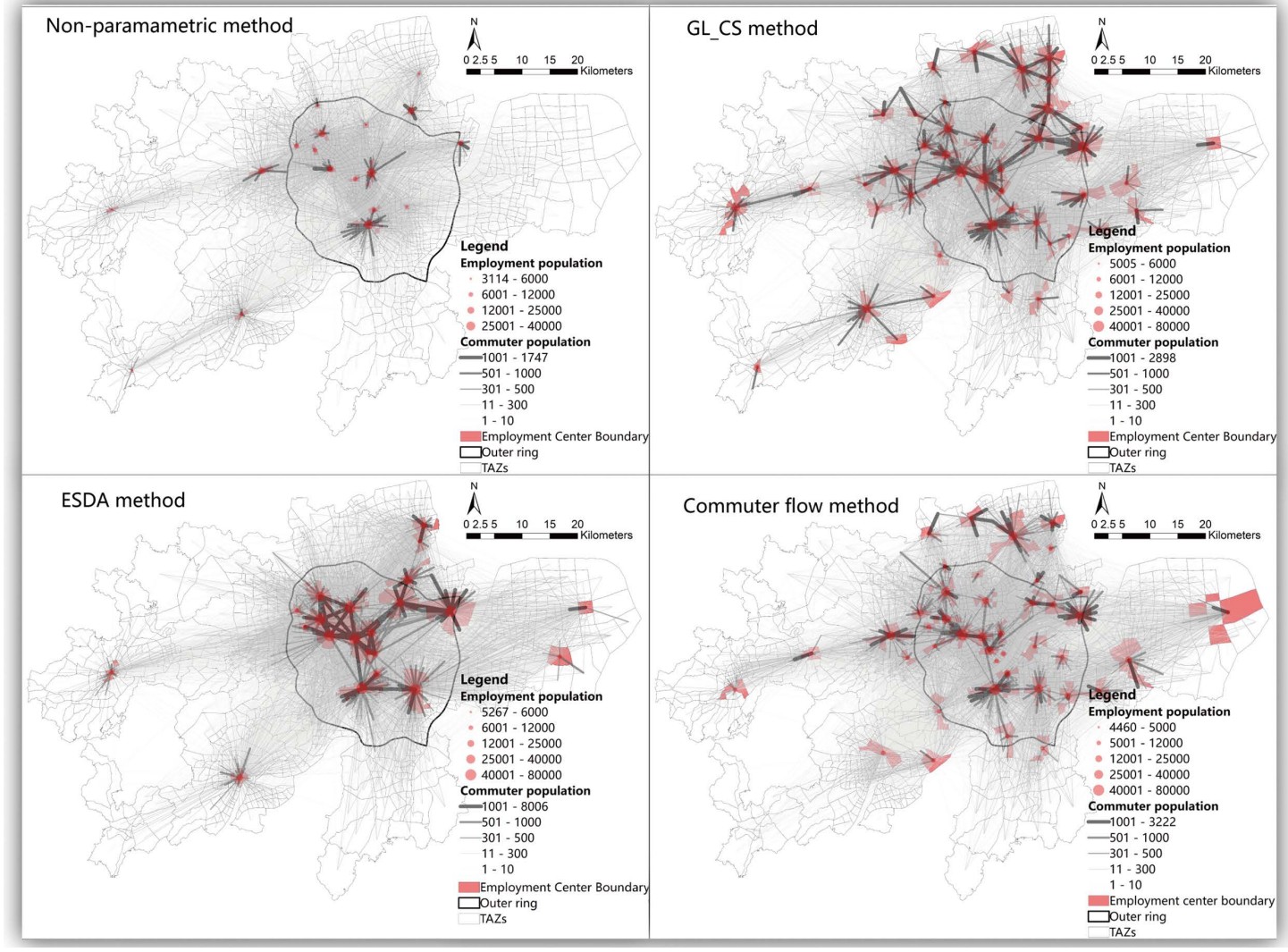

**Fig 3. The identification results of Hangzhou employment centers and the commuting flows between the centers and TAZs.** Basemap data sourced from Tianditu (https://map.tianditu.gov.cn/, Ministry of Natural Resources, Map Review No. GS(2025)1508). Administrative and TAZ boundaries are sourced from the respective municipal governments of Hangzhou, Wuhan, and Nanning, and are used with permission.

Different measurement indicators also vary in their design and application. The Primacy Index (PM) adopts the method of Bartosiewicz et al. [18]. The morphological and functional polycentricity indices are calculated as 1 minus the concentration degree of the largest center in terms of employment population and its attractiveness in commuting flows, respectively. A higher value indicates a stronger degree of polycentricity. The Entropy Index (EI) evaluates the degree of evenness in the distribution of employment among different centers. It only considers the balance of proportions without regard to the spatial location or rank order of centers [32]. The EI is highly sensitive to the number of centers—adding a new center of any size will alter its value. The Slope of the Rank–Size Distribution (SRSD) analyzes the relationship between center size and rank order, revealing the hierarchical structure of centers more effectively than the EI. The closer the absolute value of the SRSD β coefficient is to 1, the flatter the relationship becomes, indicating a higher degree of polycentricity [6,17]. However, the β coefficient is sensitive to the number of centers chosen [6,17]. To avoid bias from

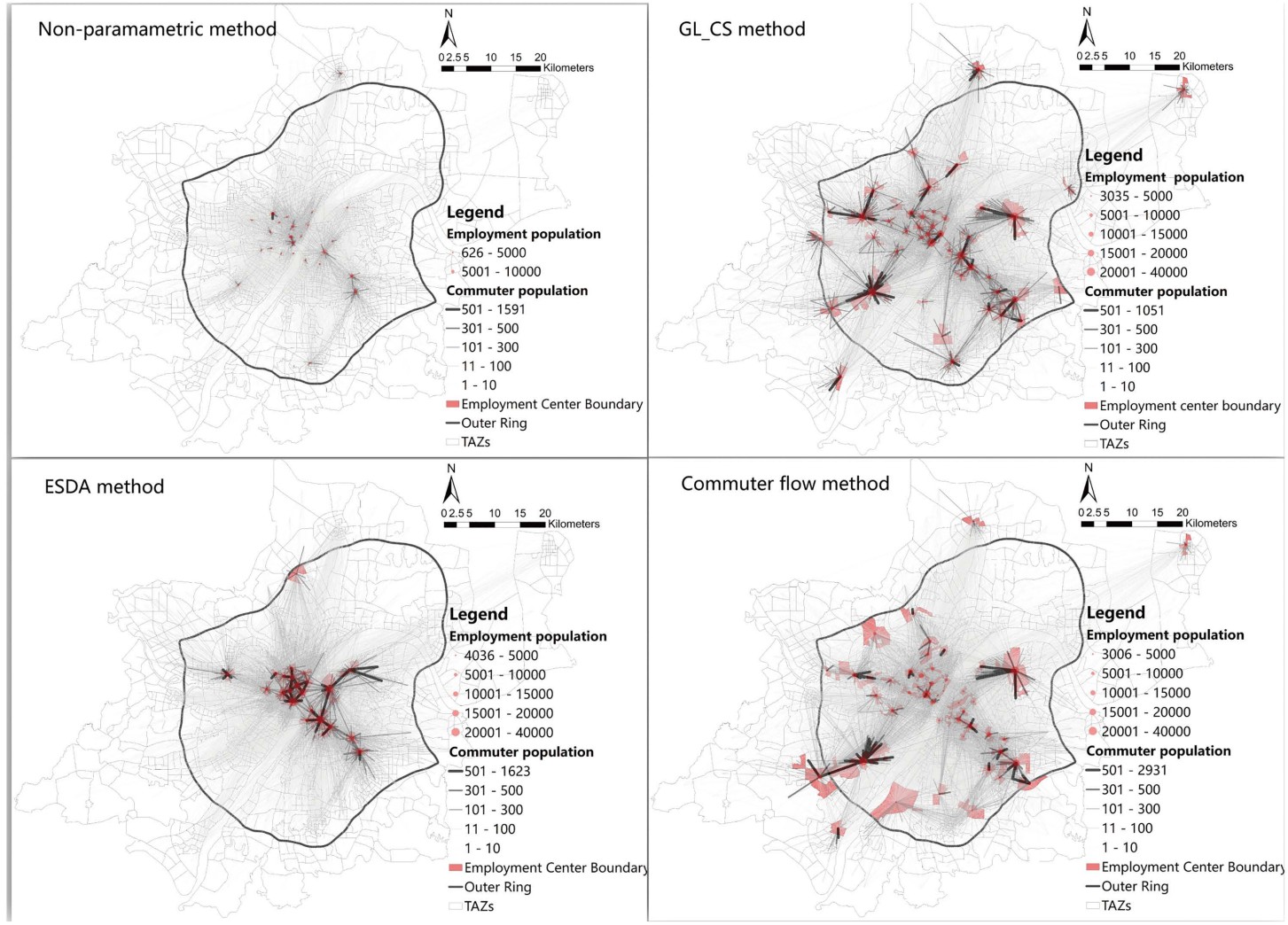

**Fig 4. The identification results of Wuhan employment centers and the commuting flows between the centers and TAZs.** Basemap data sourced from Tianditu (https://map.tianditu.gov.cn/, Ministry of Natural Resources, Map Review No. GS(2025)1508). Administrative and TAZ boundaries are sourced from the respective municipal governments of Hangzhou, Wuhan, and Nanning, and are used with permission.

selecting only a subset of centers to represent overall urban polycentricity, this study adopts the average value of all sample sizes as the final polycentricity. Network Density assesses polycentricity by comparing the actual network with an idealized network (a perfectly monocentric system) [18,19,32]. Morphological network density focuses on the evenness of employment distribution across centers, while functional network density simultaneously captures distributional balance and network connectivity—that is, whether multiple centers form a tightly linked and functionally complementary network. The Logarithmic Model directly tests the distance decay effect, examining whether employment population or connection strength decreases with increasing distance from the Central Business District (CBD). The $R^2$ value represents the goodness of fit of the monocentric model. In a fully polycentric system, the measured values of all centers approach $1/n$ (where n is the number of centers), producing a horizontal regression line. In contrast, a completely monocentric system yields an L-shaped logarithmic regression line [8].

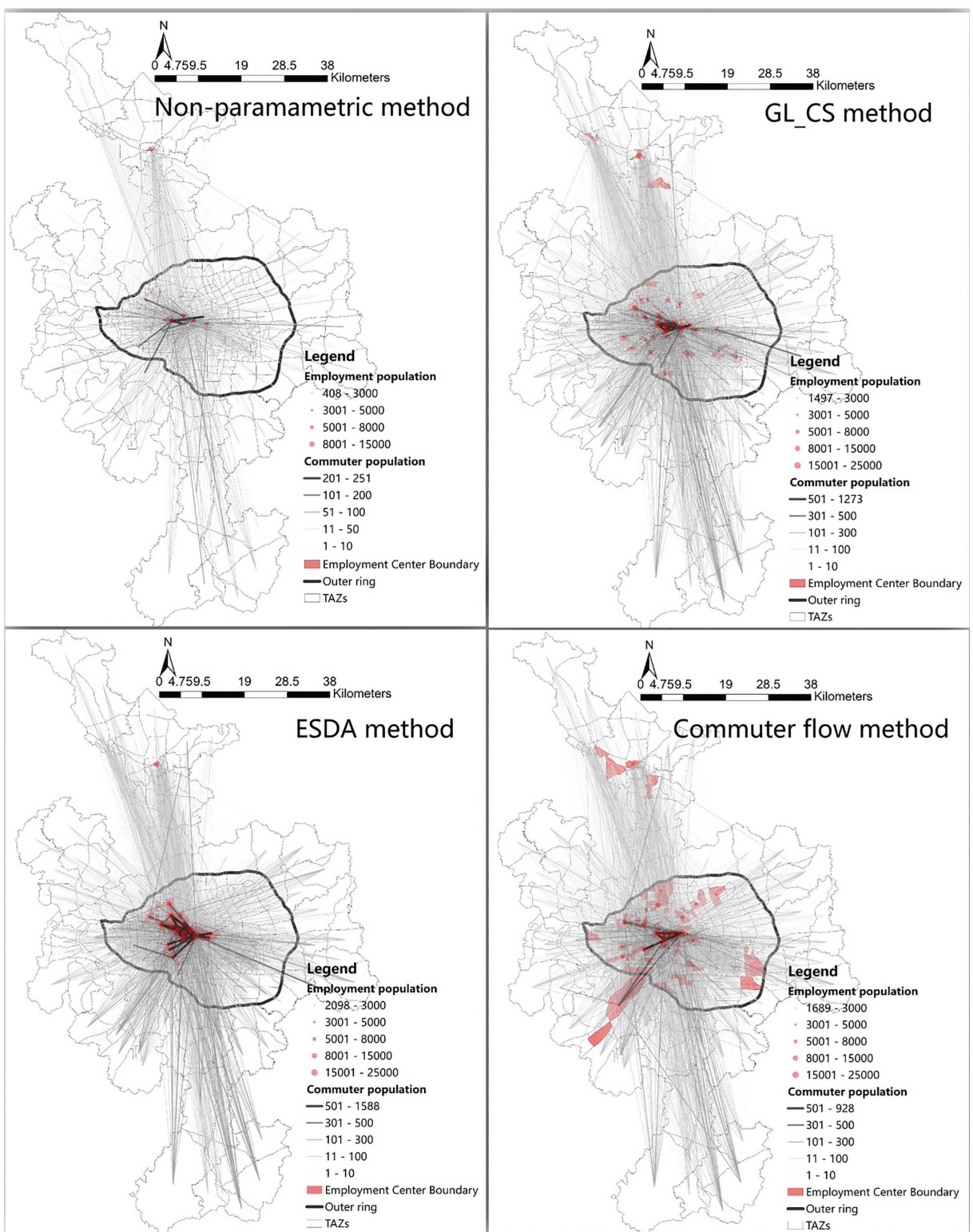

**Fig 5. The identification results of Nanning employment centers and the commuting flows between the centers and TAZs.** Basemap data sourced from Tianditu (https://map.tianditu.gov.cn/, Ministry of Natural Resources, Map Review No. GS(2025)1508). Administrative and TAZ boundaries are sourced from the respective municipal governments of Hangzhou, Wuhan, and Nanning, and are used with permission.

**Table 1. Assessment indicators for morphological and functional polycentricity.**

| Index | Dimension | Formula | |
|---|---|---|---|
| **Primacy index (PM) [18]** | Morphology | PM is defined as 1 minus the ratio of the employment population in the largest center of the urban area to the total employment population in that area | |
| | Function | PM is defined as 1 minus the ratio of the inflow of commuters traveling from urban areas to the largest center to the total inflow of commuters originating from those areas | |
| **Entropy index (EI) [18,19,32]** | Morphology | $EI = -\sum_{i=1}^{L} \frac{(z_i)\ln(z_i)}{\ln L}$ | $z_i$: the proportion of employed population in employment center i to the total number of employed population across all centers; L: the number of employment centers |
| | Function | | $z_i$: The proportion of commuters in employment center i relative to the total number of commuters across all employment centers; L: the number of employment centers |
| **The absolute value of β in SRSD model [17,29,30]** | Morphology | $\ln e_k = \alpha + \beta \ln(rank_k - 0.5)$ | $e_k$: The nodal attribute value (total employment) of employment center k; $rank_k$: The rank ordering of nodal attributes for employment center k |
| | Function | | $e_k$: The internal centrality value (in-flow commuter volume) of employment center k; $rank_k$: The rank-size ordering of the internal centrality of employment center k |
| **Network density [23,30]** | Morphology | $P_M = 1 - \frac{\sigma_M}{\sigma_{Mmax}}$ | $\delta_M$: Standard deviation of employment population in urban area; $\delta_{Mmax}$: Standard deviation of employment population in a dual-center network, where the employment population of one center is zero and the other center reaches its maximum |
| | Function | $P_F = 1 - \frac{\sigma_F}{\sigma_{Fmax}} \cdot \Delta$ <br> $\Delta = \frac{L}{L_{max}}$ | $\delta_F$: Standard deviation of commuter volume in urban area; $\delta_{Fmax}$: Standard deviation of commuter flow in a dual-center network, where the commuter volume of one center is zero and the other center reaches its maximum; $\Delta$: The network density in urban areas; L: Total commuter volume; $L_{max}$: The maximum commuter volume (e.g., the sum of the number of people working in urban areas) |
| **1-the R² value of logarithmic model [8]** | Morphology | $P_i = \alpha + \beta \ln CBD$ | $P_i$: The proportion of the employed population in an employment center I relative to the total employed population across all employment centers; $D_{CBD}$: The distance between the employment center and CBD |
| | Function | | $P_i$: The linkage strength in employment center i (the R² statistic from OLS regression); $D_{CBD}$: The distance between the employment center and CBD |

For the above indicators, the closer the value is to 1, the stronger the polycentricity.

**3.3.3. Linear mixed model (LMM).** This study analyzes the influencing factors of polycentricity measurement from three dimensions:(1) The influence of center definition methods: comparing four center identification methods with different theoretical foundations—threshold method, non-parametric method, ESDA method, and commuter flow method—to examine their differences in characterizing urban spatial structures. (2) The influence of measurement indicators: investigating the effects of five polycentricity indicators—primacy index, rank-size slope, network density, entropy index, and logarithmic model R²—on measurement outcomes from both morphological and functional dimensions. (3) The moderating role of urban development context: using Hangzhou, Wuhan, and Nanning, three cities at different stages of economic development with distinct spatial structures, as case studies to analyze whether the performance of center definition methods and measurement indicators varies with urban context. To quantify the impacts of the aforementioned factors and their interactions on polycentricity measurement, while controlling for random variations due to inter-city differences, this study constructs Linear Mixed-Effects Models (LMMs) with morphological and functional polycentricity as dependent variables, respectively. The models are specified as follows:

$$\text{Polycentricity} \sim \text{Center Definition} + \text{Measurement Indicators} + (1 \mid \text{City}) \tag{1}$$

$$\text{Polycentricity} \sim \text{Center Definition} + \text{Measurement Indicators} + \text{Center Definition} \times \text{Measurement Indicators} + (1 \mid \text{City}) \tag{2}$$

In the models, Center Definition and Measurement indicators are included as fixed-effect factors; City is included as a random intercept to capture baseline variations caused by urban heterogeneity that do not change with methods or indicators. Model 1 tests only the main effects, while Model 2 further includes the "Center Definition × Measurement indicator" interaction term to examine whether a joint effect exists. Model parameters are estimated using the Restricted Maximum Likelihood (REML) method, with the significance level set at $p < 0.001$. This modeling framework not only tests the statistical significance of fixed effects but also effectively addresses data structure issues arising from varying sample sizes or unbalanced measurement combinations across cities, while capturing within-group (different method-indicator combinations within the same city) correlation and variability.

## 4. Results and analysis

### 4.1. Comparison of morphological and functional polycentricity measurement results

#### 4.1.1. From the perspective of center identification methods: functional polycentricity is more sensitive to method choice.
By comparing the results of the four center identification methods, it is found that the measurement of functional polycentricity is much more sensitive to the choice of method than that of morphological polycentricity (Table 2, Fig 6). This suggests that the selection of identification methods can significantly influence conclusions when evaluating a city's functional connection network. Across all indicators, the boxplots for the functional dimension are generally longer and have wider whisker ranges than those for the morphological dimension. This indicates that functional polycentricity measurements show greater variability and instability. In contrast, the indicator distributions for morphological polycentricity are more concentrated, with shorter boxplots, implying greater consistency and stability in the results. In the functional dimension, the differences in indicator values obtained from different methods are substantial. For example, regarding the indicator "1 – logarithmic model R² value": In Hangzhou, all four methods measure stronger functional polycentricity than morphological polycentricity; In Wuhan, the results from the ESDA and commuting flow methods are significantly higher than those from the threshold and non-parametric methods, suggesting that these two approaches better capture the functional diffusion from the main center to the periphery, resulting in higher measured polycentricity; In Nanning, only the commuting flow method shows higher functional polycentricity, while other methods indicate stronger morphological polycentricity. Therefore, "morphological polycentricity" does not equate to "functional polycentricity." Functional polycentricity is a more complex and dynamic concept, reflecting deeper and more fluid interactions within the urban system.

#### 4.1.2. From the perspective of measurement indicators: different indicators reveal different aspects of polycentricity.
Table 3 shows that the SRSD model, the Entropy Index (EI), and the Primacy Index form a highly correlated group, exhibiting very strong negative correlations among themselves. Together, they reflect the equilibrium of the size distribution within the urban center system. Entropy and primacy indices are two sides of the same coin; their strong negative correlation jointly describes whether employment is concentrated or dispersed in space. The SRSD model reflects the rate of distance decay and shows a moderate correlation with the entropy and primacy indices, yet it cannot fully capture the complexity of the polycentric structure. The logarithmic model and the network density value form an independent group, showing very weak correlation with the aforementioned "equilibrium" group. The logarithmic model measures the spatial decay of polycentricity — the faster the decay (i.e., the higher the $1 - R^2$ value), the stronger the level of polycentricity. This indicates that the influence of secondary centers is more limited, and the spatial structure relies more heavily on the primary center. Network density, from the perspective of network relationships, measures polycentricity. A higher value indicates tighter connections between centers, reflecting a more integrated and efficient network structure. It weakly correlates with the "equilibrium" group like entropy and primacy, but shows no correlation with the SRSD or logarithmic models. Assessing urban polycentricity should involve a comprehensive comparison of indicators reflecting equilibrium, spatial decay, and network connectivity across different dimensions.

**Table 2. Comparison of polycentricity measurement indicators.**

| city | dimension | method | 1- the r-squared value of Logarithmic model | The absolute value of β in SRSD model | Entropy value(EI) | Primacy value | Network density value |
|---|---|---|---|---|---|---|---|
| Hangzhou | Morphology | GL _SC | 0.23 | 0.46 | 0.93 | 0.08 | 0.67 |
| | | Non-parametric | 0.2 | 0.55 | 0.9 | 0.18 | 0.63 |
| | | ESDA | 0.25 | 0.38 | 0.92 | 0.11 | 0.58 |
| | | Commuter flow | 0.34 | 0.48 | 0.93 | 0.08 | 0.62 |
| | Function | GL _SC | 0.36 | 0.49 | 0.92 | 0.09 | 0.76 |
| | | Non-parametric | 0.51 | 0.54 | 0.9 | 0.17 | 0.67 |
| | | ESDA | 0.69 | 0.38 | 0.92 | 0.11 | 0.71 |
| | | Commuter flow | 0.44 | 0.48 | 0.92 | 0.09 | 0.72 |
| Wuhan | Morphology | GL _SC | 0.07 | 0.42 | 0.95 | 0.08 | 0.73 |
| | | Non-parametric | 0.08 | 0.43 | 0.91 | 0.1 | 0.56 |
| | | ESDA | 0.03 | 0.39 | 0.95 | 0.11 | 0.67 |
| | | Commuter flow | 0.01 | 0.43 | 0.95 | 0.08 | 0.73 |
| | Function | GL _SC | 0.18 | 0.38 | 0.95 | 0.07 | 0.76 |
| | | Non-parametric | 0.07 | 0.43 | 0.91 | 0.1 | 0.61 |
| | | ESDA | 0.52 | 0.34 | 0.96 | 0.11 | 0.72 |
| | | Commuter flow | 0.53 | 0.41 | 0.95 | 0.07 | 0.74 |
| Nanning | Morphology | GL _SC | 0.42 | 0.54 | 0.91 | 0.12 | 0.66 |
| | | Non-parametric | 0.12 | 0.54 | 0.88 | 0.2 | 0.58 |
| | | ESDA | 0.38 | 0.36 | 0.95 | 0.12 | 0.65 |
| | | Commuter flow | 0.45 | 0.49 | 0.93 | 0.1 | 0.69 |
| | Function | GL _SC | 0.37 | 0.54 | 0.81 | 0.2 | 0.69 |
| | | Non-parametric | 0.04 | 0.55 | 0.78 | 0.25 | 0.51 |
| | | ESDA | 0.39 | 0.41 | 0.9 | 0.2 | 0.69 |
| | | Commuter flow | 0.6 | 0.48 | 0.83 | 0.15 | 0.67 |

The five measurement indicators quantify polycentricity from different perspectives. Morphological and functional polycentricity exhibit systematic differences in strength across various indicators (Table 2, Fig 6). The morphological entropy index (EI) for polycentricity is generally very high and stable (mostly above 0.9 across the three cities), indicating that employment locations are already quite spatially dispersed. However, the functional entropy values vary significantly among cities (e.g., Nanning ranges between 0.78 and 0.9), suggesting that functional balance is weaker than morphological dispersion. The Primacy Index (PM) shows minimal difference between functional and morphological polycentricity in Hangzhou and Wuhan. However, in Nanning, the functional primacy (peaking at 0.25) is significantly higher than its morphological counterpart, indicating that functional activities in Nanning are more dispersed towards sub-centers than the morphological layout. In the SRSD model, the median value of functional polycentricity is slightly higher than that of morphological polycentricity, and the boxplot difference between the two (Fig 6) is the smallest among the five indicators, making it the weakest difference between functional and morphological polycentricity. In the logarithmic model, functional polycentricity is generally higher than or close to morphological polycentricity, showing a trend in which functional polycentricity is stronger than morphological polycentricity. Network density shows that functional polycentricity is significantly higher than morphological polycentricity in almost all cities and methodologies (e.g., Hangzhou: functional dimension average ≈0.72 > morphological dimension average ≈0.63). This indicates that the functional linkages between centers within the city have formed a much tighter network than the morphological linkages. The study demonstrates that urban polycentricity cannot be judged by a single indicator. A comprehensive integration of multiple metrics is essential to grasp

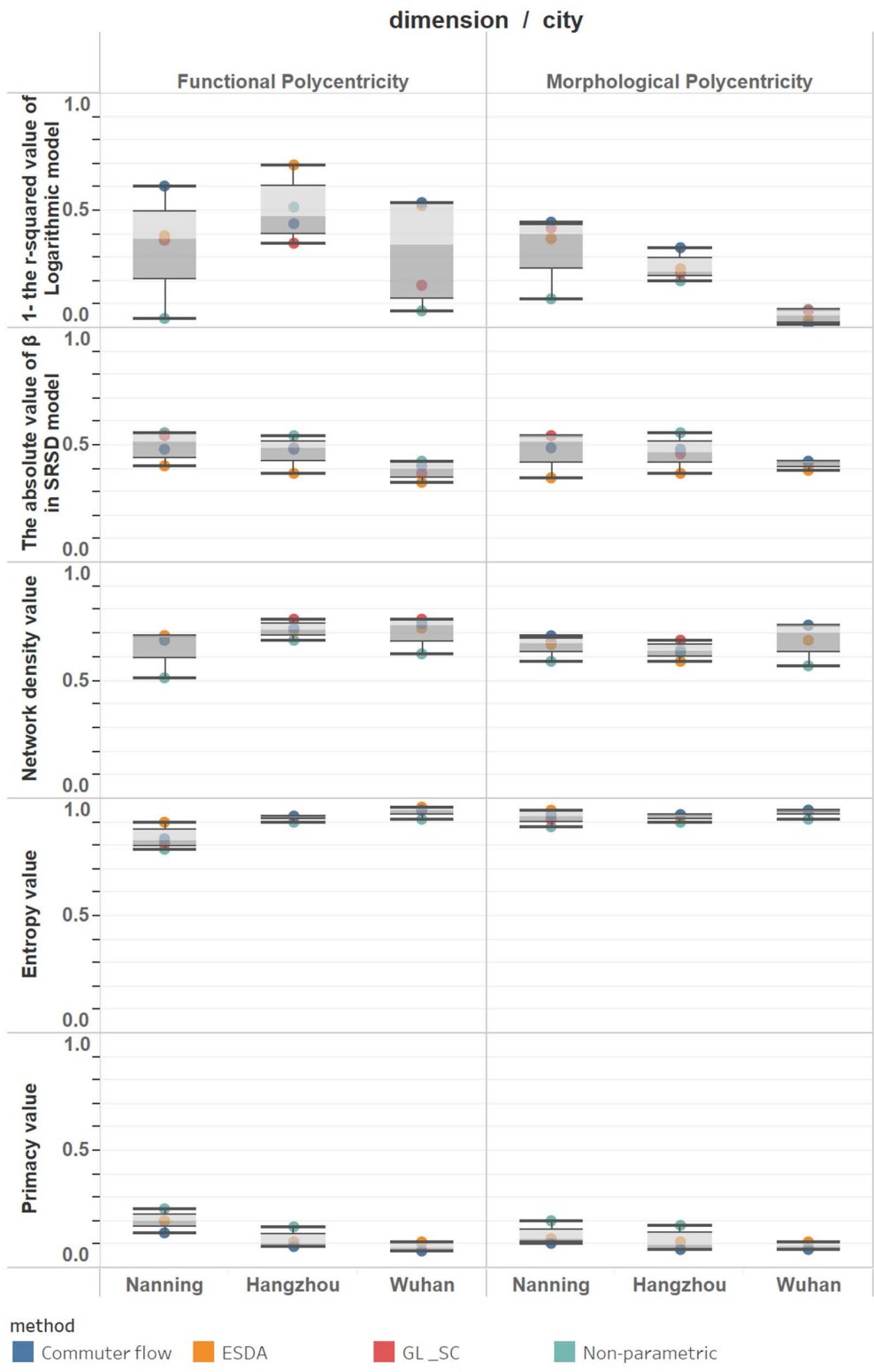

**Fig 6. Box plot of the polycentricity measure results.**

**Table 3. Pearson correlation analysis of polycentricity measurement indicators.**

| Variable | 1- the r-squared value of Logarithmic model | The absolute value of β in SRSD model | Entropy value(EI) | Primacy value | Network density value |
|---|---|---|---|---|---|
| **1- the r-squared value of Logarithmic model** | — | | | | |
| **The absolute value of β in SRSD model** | −0.053 | — | | | |
| **Entropy value(EI)** | 0.004 | −0.652*** | — | | |
| **Primacy value** | −0.038 | 0.550** | −0.808*** | — | |
| **Network density value** | 0.398 | −0.323 | 0.508* | −0.519** | — |

\* p < .05, ** p < .01, *** p < .001

its complexity. For example, while Nanning's entropy value suggests stronger morphological than functional polycentricity, its primacy index reveals the opposite trend.

**4.1.3. City comparisons: differences in polycentric structures across urban development stages.** Hangzhou demonstrates a functionally-led, mature polycentric development model. For most indicators, the median values (represented by the lines inside the boxes) for Hangzhou are markedly higher than those for the other two cities. The boxes are generally compact with short whiskers, indicating that the measured level of polycentric development is high and structurally stable across various methods, confirming Hangzhou as a mature polycentric city. The functional values for the logarithmic model are generally higher than the morphological values. Similarly, the functional network density values are slightly higher than the morphological values. This indicates that Hangzhou exhibits a functionally-led polycentric structure (See Fig 3,6). Hangzhou's polycentric development is the result of the combined effects of its unique geographical foundation and long-term, coherent planning interventions. Historically, West Lake and the Qiantang River formed natural geographical constraints. In recent years, systematic planning strategies have proactively guided the dispersal of population and industries to functional nodes such as the Future Sci-Tech City and Qianjiang New City. This planning approach, which relies on a "development corridor" framework and emphasizes functional complementarity, has effectively promoted the formation and integration of a polycentric spatial system. Consequently, the functional linkages between centers have become increasingly close-knit and networked.

Wuhan exhibits the characteristics of a dispersed and transitioning polycentric city. Although its median values for several indicators are at moderate levels, the overall distribution of results shows a downward skew, indicating that its structure is not yet stable and remains in an evolutionary stage transitioning from monocentricity to polycentricity. This transitional feature is particularly evident in the functional polycentricity measurement of the logarithmic model: its results show significant variation under different identification methods, demonstrating that methodological choices can systematically alter the diagnosis of Wuhan's functional structure—from "nearly monocentric" to "significantly polycentric." In stark contrast, the results for morphological polycentricity under different methods are generally lower and more concentrated, especially for the logarithmic model, which aligns more closely with a monocentric spatial form (see Fig 4,6). This divergence, characterized by "functional sensitivity versus morphological stability," reveals the complexity of Wuhan's spatial structure. The fundamental reason shaping this feature is Wuhan's unique "Three Towns" geographical pattern. The natural division by the Yangtze and Han Rivers has historically formed three major functional clusters: Wuchang (education/science), Hankou (commerce), and Hanyang (industry). This geographical constraint leads to a high concentration of resident travel and economic activities within each town or adjacent areas, while cross-river, long-distance, high-frequency commuting and functional linkages remain relatively limited. Consequently, in the measurement of functional polycentricity, Wuhan manifests as a network structure of "multiple coexisting but loosely connected cores." This spatial characteristic of "strong internal linkages but weak cross-regional integration" constrains the overall strength and efficiency of its functional

network, resulting in a significantly lower level of functional polycentricity compared to Hangzhou, which has a highly integrated network.

Nanning presents itself as a core-dominated, growing city, with its polycentric structure demonstrating significant instability and early-stage developmental characteristics. The results of different measurement indicators show the greatest range of variation across different identification methods (exhibiting the longest box plots), indicating that its spatial structure is extremely sensitive to methodological choices. For example, in the measurement of the logarithmic model, the non-parametric method reveals that both its morphology and function exhibit highly monocentric features, whereas the commuting flow method detects a certain tendency towards functional polycentricity. This pronounced volatility reflects that Nanning's polycentric pattern is not yet solidified and remains in a dynamic formation process (see Fig 5,6). Most indicators show that its morphological polycentricity is stronger than its functional polycentricity. This mismatch, characterized by "morphology preceding function," is rooted in the fact that its polycentricization process is primarily driven by strong policy and planning interventions. The large-scale construction of new districts in the short term has rapidly expanded the urban spatial framework, shaping a polycentric pattern dominated by "morphological expansion." However, the city's economic flows, commuting patterns, and social networks still require time to develop and restructure. The traditional main center (such as Qingxiu District) continues to hold a dominant functional position, and newly established clusters remain deeply dependent on it in terms of functionality. Consequently, this leads to a noticeable disparity between morphological and functional polycentricity in the measurements.

This study reveals a widespread phenomenon of non-linear disjunction between "morphological polycentricity" and "functional polycentricity" during the spatial transformation of Chinese cities. Integrating the distinct spatial structures, planning trajectories, and developmental stages of three representative cities, the research demonstrates that the systematic differences observed in polycentricity measurement results are, in essence, a systematic and quantitative reflection of each city's unique geographical foundations, the temporal logic of planning interventions, and the actual integration level of functional networks within their spatial structures.

## 4.2. Analysis of factors influencing polycentricity measurement

### 4.2.1. Fixed-effects of factors influencing polycentricity.
The analysis results in Table 4 show that: Firstly, the main effects of polycentricity measurement indicators are extremely significant. For both morphological and functional polycentricity, the measurement indicators are the decisive factor causing differences in measurement results ($p < .000$). This confirms that different indicators measure polycentricity from different dimensions, their values are not directly comparable, and the choice of evaluation indicators has a substantial impact on the measurement outcomes of morphological and functional polycentricity. Secondly, the main effects of the center identification methods are not significant in the morphological polycentricity models (both non-interactive and interactive) and in the functional polycentricity non-interactive model ($p > 0.05$). However, they become significant in the functional polycentricity interactive model ($p = 0.041$). This indicates that center identification methods do not systematically alter the level of morphological polycentricity but do systematically affect the measurement results of functional polycentricity. Thirdly, there is no significant interaction effect between identification methods and measurement indicators in the morphological polycentricity model ($p = 0.206$), suggesting that different identification methods perform similarly across various indicators and that the measurement patterns for morphological polycentricity are stable. In contrast, a significant method × indicator interaction effect appears in the functional polycentricity model ($p < .000$), indicating that different identification methods yield inconsistent results across different indicators. This inconsistency highlights the complexity of measuring functional polycentricity, which is more sensitive to method selection than morphological polycentricity. Fourthly, the AIC and BIC values of the non-interactive morphological polycentricity model (−117.1 and −96.2, respectively) are better than those of the interactive model (−107.7 and −61.6), indicating that the non-interactive model fits the data better while using fewer

**Table 4. Restricted maximum likelihood parameter estimates and fitting statistics of indicators in LMMs for the interaction effect between centers, index on polycentricity.**

| Dimension | method | Effect | df | F | p-value | log Lik | AIC | BIC |
|---|---|---|---|---|---|---|---|---|
| Morphological polycentricity | Non-interactive | Intercept | 1 | 2239.901 | <.000 | −137.138 | −117.138 | −96.195 |
| | | Centers | 3 | 0.709 | 0.550 | | | |
| | | Index | 4 | 217.487 | <.000 | | | |
| | Interactive | Intercept | 1 | 2853.544 | <.000 | −151.666 | −107.666 | −61.590 |
| | | Centers | 3 | 0.904 | 0.445 | | | |
| | | Index | 4 | 277.070 | <.000 | | | |
| | | Centers* Index | 12 | 1.370 | 0.206 | | | |
| Functional polycentricity | Non-interactive | Intercept | 1 | 1590.553 | <.000 | −106.631 | −86.631 | −65.687 |
| | | Centers | 3 | 1.613 | 0.196 | | | |
| | | Index | 4 | 102.682 | <.000 | | | |
| | Interactive | Intercept | 1 | 2882.696 | <.000 | −142.309 | −98.309 | −52.234 |
| | | Centers | 3 | 2.923 | 0.041 | | | |
| | | Index | 4 | 186.099 | <.000 | | | |
| | | Centers* Index | 12 | 4.062 | <.000 | | | |

parameters. For functional polycentricity, the interactive model includes significant interaction terms. Although its AIC/BIC values appear larger, this is because it contains more necessary parameters to capture the complex relationships. Therefore, when constructing LMM models for morphological polycentricity, a simple model considering only measurement indicators is sufficient. For functional polycentricity, however, the interaction between center identification methods and measurement indicators must be considered; otherwise, the model specification would be incorrect, potentially leading to biased conclusions.

**4.2.2. Random effects of factors affecting polycentricity.** The analysis results in Table 5 show that: Firstly, the model successfully captures the main sources of variation. The variance of repeated measurements in all models is statistically highly significant (Wald Z > 4.4, p < .0001), indicating that the variation resulting from multiple measurements of the same city using different methods and indicators constitutes the primary component that the model needs to explain. Different measurement approaches can produce substantially different results. Secondly, the inherent differences between cities are not significant. The estimated variance between cities in all models is 0.00000 (with standard errors also equal to 0), which means that after controlling for the effects of methods and indicators, there are no significant inherent differences in polycentricity levels among Hangzhou, Wuhan, and Nanning. The observed differences between cities may be a superficial effect caused by differing measurement approaches rather than a fundamental distinction. Essentially, all inter-city differences can be explained by variations in measurement methods and evaluation indicators. Thirdly, the data variability of functional polycentricity is greater than that of morphological polycentricity. The estimated variance of repeated measurements for functional polycentricity is higher than that for morphological polycentricity, indicating that functional polycentricity is more sensitive and uncertain in measurement. Fourthly, introducing interaction terms improves model fit. In both the morphological and functional dimensions, models including interaction terms have lower estimated variances for repeated measurements compared with non-interactive models, indicating that the interaction terms account for a portion of the variation that would otherwise be attributed to random error, thereby improving model performance. This study demonstrates that how polycentricity is measured (method + indicator) has a greater impact on the results than which city is being measured. When comparing different cities, methodological differences must be carefully considered. From a statistical perspective, during the process of Chinese large cities evolving toward polycentric structures, the

**Table 5. Covariance Parameters estimates and fitting statistics of random effects in LMM model.**

| Dimension | method | Parameter | Estaimate | Std. Error | Wald Z | Sig. |
|---|---|---|---|---|---|---|
| Morphological Polycentricity | Non-interactive | Repeated Measures Variance | 0.005955 | 0.001087 | 5.477 | <.000 |
| | | City Variance | 0.00000[b] | 0.00000 | . | . |
| | Interactive | Repeated Measures Variance | 0.004674 | 0.000853 | 5.477 | <.000 |
| | | City Variance | 0.00000[b] | 0.00000 | . | . |
| Functional Polycentricity | Non-interactive | Repeated Measures Variance | 0.009902 | 0.001808 | 5.477 | <.000 |
| | | City Variance | 0.00000[b] | 0.00000 | . | . |
| | Interactive | Repeated Measures Variance | 0.005463 | 0.000997 | 5.477 | <.000 |
| | | City Variance | 0.00000[b] | 0.00000 | . | . |

b: This covariance parameter is redundant. The test statistics and confidence intervals cannot be calculated.

common challenges in their structure (such as method sensitivity revealed by measurement) may outweigh the individual differences between cities.

**4.2.3. LMM analysis of morphological and functional polycentricity.** The LMM model for morphological polycentricity did not include interaction terms (Table 6). The study found that measurement indicators are the primary factor determining the level of morphological polycentricity, while the choice of center identification method has no significant effect on the results. Firstly, the non-parametric method and network density indicator were set as the reference groups for method and indicator, respectively (estimates set to 0 for comparison). The intercept estimate is 0.633 and highly significant (p<.001), indicating that when using the non-parametric method and the network density indicator, the predicted average level of morphological polycentricity is 0.633. Secondly, the p-values for the three identification methods are all much greater than 0.05, suggesting that after controlling for the effect of measurement indicators, these methods do not show systematic or significant differences compared with the non-parametric method in measuring morphological polycentricity. Thirdly, other measurement indicators show large and highly significant differences (p<.001) compared with the reference "network density." This indicates that different measurement indicators have a decisive impact on the results. The estimate for the EI indicator is positive (+0.278), meaning that under the same identification method, morphological polycentricity measured by EI is significantly higher than that measured by network density. Estimates for

**Table 6. Fixed effect estimation and fitting statistics of indicators in LMMs for the interaction effect between centers, index on morphological polycentricity.**

| Parameter | Estaimate | Std.Error | df | t | Sig. |
|---|---|---|---|---|---|
| Intercept | .633333 | .028178 | 60 | 22.476 | .000 |
| GL_CS | .027333 | .028178 | 60 | .970 | .336 |
| ESDA | −.000667 | .028178 | 60 | −.024 | .981 |
| Commuter flow | .030000 | .028178 | 60 | 1.065 | .291 |
| Non-parametric | 0[b] | 0 | . | . | . |
| Logarithmic model | −.432500 | .031504 | 60 | −13.728 | .000 |
| SRSD | −.191667 | .031504 | 60 | −6.084 | .000 |
| EI | .278333 | .031504 | 60 | 8.835 | .000 |
| PM | −.534167 | .031504 | 60 | −16.955 | .000 |
| Network density | 0[b] | 0 | . | . | . |

b: This parameter is set to zero because it is redundant.

the Logarithmic model, SRSD, and PM indicators are negative (−0.433, −0.192, −0.534), indicating that morphological polycentricity measured using these indicators is significantly lower than that measured by network density. The findings demonstrate that in studies of morphological polycentricity, how polycentricity is measured is more important than which center identification method is used. Comparisons across studies must be based on the same indicators; otherwise, the comparisons are meaningless. Urban planners assessing whether a city has achieved a polycentric structure should ideally use multiple indicators to comprehensively capture different dimensions of morphological polycentricity. For example, using the EI indicator might suggest that the polycentricity goal has been achieved, while using the SRSD indicator might suggest it has not.

In contrast to the morphological polycentricity model, the measurement results of functional polycentricity are jointly influenced by the center identification method, measurement indicators, and the strong interaction between them (Table 7).

**Table 7. Fixed effect estimation and fitting statistics of indicators in LMMs for the interaction effect between centers, index on functional polycentricity.**

| Parameter | Estaimate | Std.Error | df | t | Sig. |
|---|---|---|---|---|---|
| Intercept | .596667 | .042674 | 60 | 13.982 | .000 |
| GL_CS | .140000 | .060351 | 60 | 2.320 | .024 |
| ESDA | .110000 | .060351 | 60 | 1.823 | .073 |
| Commuter flow | .113333 | .060351 | 60 | 1.878 | .065 |
| Non-parametric | 0[b] | 0 | 60 | . | . |
| Logarithmic model | −.390000 | .060351 | 60 | −6.462 | .000 |
| SRSD | −.090000 | .060351 | 60 | −1.491 | .141 |
| EI | .266667 | .060351 | 60 | 4.419 | .000 |
| PM | −.423333 | .060351 | 60 | −7.015 | .000 |
| Network density | 0[b] | 0 | 60 | . | . |
| [GL_CS] * [Logarithmic model] | −.043333 | .085349 | 60 | −.508 | .614 |
| [GL_CS] * [SRSD] | −.176667 | .085349 | 60 | −2.070 | .043 |
| [GL_CS] * [EI] | −.110000 | .085349 | 60 | −1.289 | .202 |
| [GL_CS] * [PM] | −.193333 | .085349 | 60 | −2.265 | .027 |
| [GL_CS] * [Network density] | 0[b] | 0 | 60 | . | . |
| [ESDA] * [Logarithmic model] | .216667 | .085349 | 60 | 2.539 | .014 |
| [ESDA] * [SRSD] | −.240000 | .085349 | 60 | −2.812 | .007 |
| [ESDA] * [EI] | −.046667 | .085349 | 60 | −.547 | .587 |
| [ESDA] * [PM] | −.143333 | .085349 | 60 | −1.679 | .098 |
| [ESDA] * [Network density] | 0[b] | 0 | 60 | . | . |
| [Commuter flow] * [Logarithmic model] | .203333 | .085349 | 60 | 2.382 | .020 |
| [Commuter flow] * [SRSD] | −.163333 | .085349 | 60 | −1.914 | .060 |
| [Commuter flow] * [EI] | −.076667 | .085349 | 60 | −.898 | .373 |
| [Commuter flow S] * [PM] | −.183333 | .085349 | 60 | −2.148 | .036 |
| [Commuter flow] * [Network density] | 0[b] | 0 | . | . | . |
| [Non-parametric] * [Logarithmic model] | 0[b] | 0 | . | . | . |
| [Non-parametric] * [SRSD] | 0[b] | 0 | . | . | . |
| [Non-parametric] * [EI] | 0[b] | 0 | . | . | . |
| [Non-parametric] * [PM] | 0[b] | 0 | . | . | . |
| [Non-parametric] * [Network density] | 0[b] | 0 | . | . | . |

b: This parameter is set to zero because it is redundant.

Firstly, the model uses the non-parametric method and network density as reference points. The intercept of 0.597 (p < .001) indicates that when using the non-parametric method and network density indicator, the predicted average level of functional polycentricity is 0.597. Secondly, the GL_CS method has an estimate of 0.140, which is statistically significant (p = 0.024), indicating that, after controlling for the effect of indicators, the overall functional polycentricity measured using GL_CS is significantly higher than that measured by the non-parametric method. The ESDA and commuter flow methods are not significant, suggesting that their average differences compared with the non-parametric method are statistically uncertain. Thirdly, the EI indicator has an estimate of +0.267 and is highly significant (p < .001), meaning that under the same method, functional polycentricity measured by EI is significantly higher than that measured by network density. The Logarithmic model and PM indicators have estimates of −0.390 and −0.423, respectively, both highly significant (p < .001), indicating that results measured using these two indicators are significantly lower than those measured by network density. The SRSD estimate is negative but not significant (p = 0.141), so its effect is uncertain. Fourthly, the GL_CS × PM interaction term has an estimate of −0.193 and is significant (p = 0.027), indicating that when using the GL_CS method, functional polycentricity measured by the PM indicator is significantly lower than the reference scenario (non-parametric method × network density). The ESDA × Logarithmic model interaction term has an estimate of +0.217 and is significant (p = 0.014), showing that when using the ESDA method, functional polycentricity measured by the Logarithmic model indicator is significantly higher. Other significant interactions, such as [GL_CS] × [SRSD], [ESDA] × [SRSD], [Commuter flow] × [Logarithmic model], and [Commuter flow] × [PM], also reach significance, demonstrating that interaction effects are widespread rather than isolated phenomena. The findings indicate that indicator sensitivity varies by method, and the measurement results of functional polycentricity are highly dependent on the specific "method–indicator" combination. There is no universally optimal approach. Comparisons of functional polycentricity results across studies must be made with extreme caution, and only comparisons using exactly the same method–indicator combinations are meaningful. In functional polycentricity research, it is recommended to test different method–indicator combinations as a necessary step in sensitivity analysis to verify the robustness of study conclusions.

## 5. Discussion

### 5.1. A practical guidance framework for measuring urban polycentricity: comprehensive recommendations based on LMM results

This study, through linear mixed models, reveals that the identification methods for centers and the measurement indicators have a complex and nonlinear impact mechanism on the assessment results of both morphological and functional urban polycentricity. This mechanism manifests primarily at two levels: the definition of "centers" and the measurement of "polycentricity."

Firstly, the definition of "centers" serves as the logical starting point of measurement. Different methods define "center" based on varying criteria, which may lead to differences in the number, location, and range of centers identified. The LMM analysis indicates that morphological polycentricity represents a relatively stable and highly consensual "static pattern," which is not sensitive to the choice of center definition method. The variance component analysis shows that a small repeated measurement error, suggesting that the spatial distribution of urban physical space serves as a relatively objective benchmark. Functional polycentricity, on the other hand, is portrayed as a complex, dynamic "interactive process," with the repeated measurement variance for functional polycentricity being much higher than for the morphological dimension. This means that using different center definition methods for the same city can lead to vastly different conclusions. The fixed effects model further reveals that in the functional dimension, both the main effect of center identification methods (Centers, p = 0.041) and their interaction effect with measurement indicators (Centers * Index, p < .000) are significant. Therefore, the way in which centers are defined fundamentally frames the functional structure observed. When comparing functional polycentricity, the method of defining centers must be clearly stated as a core context. Secondly, the choice of measurement indicator is a decisive factor influencing the results. For example, the entropy index (EI), which measures distribution balance, has a significantly positive effect on polycentricity (functional dimension: +0.267), while the primacy

index (PM), which measures the dominance of the primary center, has a significantly negative effect (functional dimension: −0.423). This finding aligns with the conclusion of Bartosiewicz and Marcińczak [18] regarding systematic differences among various measurement indicators. More importantly, the significant "method-indicator" interaction effect observed in the functional dimension indicates that its measurement outcomes are jointly determined by specific "method-indicator" combinations. A particular center identification method may systematically enhance or diminish the sensitivity of a specific indicator. For instance, combining the ESDA method with the logarithmic model significantly increases the measurement value, while combining the GL_CS method with the primacy index significantly decreases it. Given the high dependence of the results on the "method-indicator" combination, conclusions drawn from a single combination may not be robust.

To translate the empirical findings of this study into a practical tool directly applicable to researchers and urban planning decision-makers, this research constructs a comprehensive methodological selection framework (Table 8). This framework aims to guide users in selecting appropriate combinations of methods and indicators based on specific research objectives, data conditions, and urban development contexts, thereby enhancing the reliability, validity, and comparability of polycentricity measurements. The core principles guiding the design of the framework are as follows: First, it is goal-oriented. Different measurement indicators essentially serve as proxy variables for distinct theoretical dimensions of urban spatial structure (e.g., EI measures the evenness of size distribution, SRSD measures spatial dependency patterns, and network density measures the strength of connections between centers). Second, it strictly distinguishes between morphological and functional dimensions. These two dimensions not only differ in their conceptual foundations but also exhibit systematic differences in the stability of measurement results and their sensitivity to methodological choices, necessitating

**Table 8. Comprehensive framework for selecting methods to measure urban polycentricity.**

| Measurement objective | Recommended Indicators | Recommended center identification methods | Core Rationale and Precautions |
|---|---|---|---|
| Evaluate scale distribution balance | EI or PM | -**Morphological Dimension**: All four methods are acceptable.<br>-**Functional Dimension**: Results from all four methods are relatively stable; non-parametric methods or GL_CS are preferred. | - **Morphology**: Indicators (EI/PM) are decisive factors, method choice does not affect conclusions.<br>- **Function**: There is a slight negative interaction between GL_CS and PM (−0.193*); the same indicator must be used, otherwise results are incomparable. |
| Analyze spatial dependence and distance decay | SRSD, Logarithmic model | - **Morphological Dimension**: All four methods are acceptable.<br>- **Functional Dimension**: Methods must be selected with great caution; avoid combining ESDA or Commuting Flow methods with SRSD and Commuting Flow methods with Logarithmic R² (significant negative interaction). Non-parametric methods or GL_CS can be considered with these indicators, but note that GL_CS also has a negative interaction with SRSD. | This is the domain with the strongest "method–indicator" interaction effect. Incorrect method combinations (e.g., ESDA+SRSD) systematically underestimate functional polycentricity. In the morphological dimension, polycentricity levels measured by these indicators are significantly lower than those from Network Density or EI. |
| Evaluate network connection strength | Network density | - **Morphological Dimension**: All four methods are acceptable.<br>- **Functional Dimension**: Non-parametric methods (as a baseline) or GL_CS methods (with a significant positive main effect, +0.140*) are recommended. | - **Morphology**: Network Density is a stable benchmark, but its measured polycentricity is significantly lower than EI and significantly higher than SRSD, Logarithmic R², and PM.<br>- **Function**: GL_CS yields higher estimates of network connection strength. |
| Comprehensive diagnosis and sensitivity analysis | Multi-indicator Combination (EI, Network density, SRSD, Logarithmic model | - **Morphological Dimension**: Any one method can be used, but run all or key indicators simultaneously.<br>- **Functional Dimension**: Multi-method–multi-indicator combination tests are mandatory; use non-parametric methods as a baseline and GL_CS to test the impact on Network density, and use ESDA/Commuting Flow to test results—but avoid the problematic combinations of SRSD and R². | - **Morphology**:: A single indicator or method may mislead; combining them fully captures different dimensions of polycentricity (balance, connectivity, spatiality). - **Functional**: Sensitivity analysis is essential to reveal whether conclusions depend on specific method combinations. |

distinct strategies. Finally, it emphasizes the practical implications of interaction effects. Particularly for functional polycentricity, failing to account for systematic biases in specific "method-indicator" combinations (e.g., avoiding the use of ESDA/Commuter Flow with SRSD/R² for spatial-functional analysis due to strong negative interactions) may lead to misleading conclusions about the structure of urban functional networks. Therefore, a scientific understanding and practical assessment of urban polycentricity should shift from a paradigm of "seeking the single truth" to one of "understanding and managing the sources of uncertainty." This framework serves as a practical tool for this paradigm shift. It requires researchers to explicitly articulate their methodological pathways (the methods and indicators used) and conduct sensitivity analyses. For planners, the framework highlights that the simple label of "polycentricity" conceals a complex spatial reality, and scientific decision-making should be based on a comprehensive evaluation of results across different measurement dimensions. To judge if "polycentricity" goals are met, compare results across indicators. For example, if EI indicates success but SRSD indicates failure, the city may be balanced in scale distribution but still rely on a dominant center spatially.

### 5.2. Case diagnostics: unpacking morpho-functional divergence through urban trajectories

The variance component analysis of the LMM revealed that the estimated variance among the three case cities was zero. This indicates, on one hand, that the limited sample size in this study may be a contributing factor. On the other hand, it also suggests that the differences in polycentricity measurements are primarily attributable to the uncertainties inherent in the measurement process itself, rather than to the inherent attributes of the cities. For example, the large repeated measurement variance of functional polycentricity (0.009902 in the non-interactive model) suggests that the internal variability resulting from applying different methods and indicators to the same city may even exceed the actual differences that exist between different cities. Therefore, when conducting intercity comparisons, if the substantial uncertainty introduced by measurement methods is ignored and conclusions from different studies (which may have used different method combinations) are compared directly, the reliability of such comparisons becomes questionable. For instance, a city A's functional polycentricity value measured using the "commuting flow + network density" combination may be higher than that of city B measured using the "non-parametric + primacy index" combination. However, to what extent this difference reflects genuine spatial structural variation—and to what extent it stems from methodological effects—cannot be determined. Hence, future comparative urban studies must standardize methodological frameworks or incorporate sensitivity analyses as a prerequisite. The LMM model provides an effective tool for quantifying such measurement uncertainty. Therefore, the application of the urban polycentricity measurement framework (Table 8) must be adapted to the specific developmental stage and spatial structural characteristics of each city. For mature polycentric cities (e.g., Hangzhou), most method-indicator combinations within the framework should yield relatively consistent results, as functional and morphological polycentricity tend to be synergistic. For cities with transitional structures (e.g., Wuhan), emphasis should be placed on the functional dimension of the framework. Multi-combination analysis should be used to reveal the divergence between a "morphologically monocentric yet functionally networked" structure, and the results of spatial dependency indicators should be interpreted with caution. For policy-driven emerging polycentric cities (e.g., Nanning), the analytical focus should be on comparing the gaps between morphological and functional indicators, and employing sensitivity analysis in the functional dimension to understand the causes of high score variability and network instability, thereby evaluating the actual effectiveness of planning implementation.

### 5.3. Policy sensitivity and planning risks: from methodological bias to strategic missteps

This study finds that the measurement outcomes of urban polycentricity are highly dependent on specific "method-indicator" combinations. This methodological dependency is far from a purely academic discussion; its deeper implication lies in revealing a critical risk in urban planning practice: spatial diagnoses based on biased or single methods can systematically distort the understanding of a city's true structure, thereby inducing strategic planning errors and severe

misallocation of public resources. In the context of China establishing "polycentric development" as a top-level spatial strategy, understanding and avoiding such risks holds urgent practical significance.

Methodological choices directly influence the perception of urban spatial structure. Taking Wuhan as an example, if the "commuter flow method," sensitive to dynamic flows, is combined with the "logarithmic model," which measures spatial decay, its functional polycentric characteristics are highlighted. This diagnosis naturally leads to policies strengthening cross-river transport integration and promoting functional synergy among the "Three Towns" under a "networked governance" approach. Conversely, if the "threshold method," based on static employment distribution, is combined with the "primacy index (PM)," which measures dominance, it reinforces the morphological perception of an "absolutely dominant main center with weak secondary centers." This, in turn, supports the traditional central place system strategy of alleviating pressure on the main city by cultivating secondary "growth poles" like the Optics Valley. Both stem from data, yet lead to vastly different intervention logics due to differing methodological paths. In the case of Nanning, if decision-makers rely solely on morphological indicators showing high "entropy values," they may prematurely conclude that the polycentric structure is mature, thus dispersing major infrastructure evenly across various new districts. However, functional indicators like "network density" and the "primacy index" reveal that urban functions remain highly dependent on the Qingxiu main center. Blindly dispersing investments may lead to insufficient vitality in new districts, excessive congestion in the main city, and difficulty in forming synergistic effects among new districts due to a lack of effective interconnections.

Spatial misdiagnoses caused by methodological bias can trigger chain reactions in the planning implementation phase. In cities like Nanning, characterized by "morphology preceding function," relying on morphological polycentricity conclusions to prematurely plan large transport hubs or public service facilities may face challenges of low utilization rates and high operational costs. In Wuhan, underestimating its potential for functional networking may result in insufficient investment in key facilities that promote linkages between "multiple cores," such as cross-river passages and ring expressways. In cities like Hangzhou with mature functional networks, policies should focus on enhancing network resilience and functional upgrading. Mistakenly persisting with the discrete strategy of cultivating secondary growth poles may lead to redundant construction and internal homogeneous competition. Conversely, in cities like Nanning where secondary center functions are still fragile, prematurely promoting mixed land use or introducing high-end industries may fail due to a lack of local demand and supporting networks. If higher-level government assessments of local "polycentric" strategies rely solely on morphological conclusions derived from a single simple method (e.g., the density threshold method based on administrative units), they will be unable to accurately measure the actual effectiveness of functional integration. This may even perversely incentivize localities to pursue short-term behaviors focused on "spatial expansion over functional integration," contrary to the policy's original intent.

Therefore, this study advocates placing methodological reflection at the forefront of the planning decision-making process. Important spatial strategic plans and environmental impact assessment reports must explicitly state the center identification methods and measurement indicators used to evaluate urban spatial structure (especially functional structure), and explain the theoretical basis and applicability boundaries of their choices. This should become a fundamental standard for planning scientific rigor and credibility. Planning analysis should abandon the old paradigm of seeking a single definitive conclusion. It is recommended to adopt the practical framework constructed in this study (Table 8) for cross-validation and sensitivity analysis using multiple method-indicator combinations, presenting a set of possible structural diagnoses (e.g., "from strong monocentricity to weak polycentricity"). The range of uncertainty in the conclusions itself should be transformed into key information for planning decisions. Major infrastructure projects must undergo "stress testing" under different structural assumptions to assess the robustness of their plans. A dynamic evaluation and learning mechanism should be established. Urban spatial structure is in dynamic evolution, especially in transitioning and emerging cities like Wuhan and Nanning. The assessment of planning implementation effectiveness should be an ongoing process, regularly monitoring changes in the synergy between "morphology and function" using multi-source flow data and multi-dimensional indicators. This allows for timely adjustments to planning strategies, shifting from static "blueprint planning" to dynamic "process intervention."

In summary, urban planning is not only the art of space but also an evidence-based scientific decision-making process. This study demonstrates that the measurement of the core concept of "polycentricity" itself is fraught with methodological "traps." Acknowledging and managing this uncertainty, shifting from pursuing a "single definitive truth" to understanding "multiple possible scenarios," is an inevitable choice for avoiding planning risks and enhancing strategic resilience. Only spatial diagnoses built upon methodological introspection can provide a reliable foundation for moving towards a more just, efficient, and sustainable urban future.

## 6. Conclusion

This study comprehensively explored the influencing factors of polycentricity measurement from three perspectives: center identification methods, measurement indicators, and the morphological–functional distinction. Using the linear mixed model (LMM) and multi-indicator cross-validation, it systematically assessed how the definition of centers and the choice of indicators affect the measurement results of polycentricity, and compared the morphological and functional polycentricity of Hangzhou, Wuhan, and Nanning. The findings reveal several key points. Firstly, how centers are defined determines what kind of structure can be observed. Functional polycentricity is highly sensitive to the method of center identification, whereas morphological polycentricity is relatively robust. Secondly, measurement indicators are the primary determinants of the numerical level of polycentricity. In the functional dimension in particular, the choice of indicators and the method of defining centers are tightly intertwined; thus, it is meaningless to discuss indicator values in isolation. The "method–indicator" combination must be treated as an integrated analytical unit. Thirdly, morphological polycentricity represents a relatively stable "static configuration," while functional polycentricity reflects a complex "dynamic interaction process." The two follow distinct logics; therefore, paradigms developed for morphological polycentricity cannot be simply applied to the functional dimension. Functional measurements inherently contain a degree of uncertainty. Finally, Hangzhou, Wuhan, and Nanning represent different stages of polycentric evolution, confirming the existence of a transitional path from morphological monocentricity to functional polycentricity. This demonstrates that research on polycentricity requires a dynamic and developmental perspective.

Therefore, when analyzing urban polycentricity—particularly functional polycentricity—the methodological choices (how centers are defined and which indicators are selected) fundamentally shape the research conclusions themselves. The sensitivity of center definition implies that adopting a commuting-flow-based approach versus a GL_CS approach may yield drastically different assessments of the same city's functional polycentricity. The strong interaction effects between measurement indicators and center definition methods further indicate that indicator performance is highly contingent on the method used. In essence, polycentricity measurement is the joint outcome of specific "method–indicator" combinations. At the policy practice level, diagnostic conclusions drawn from different "method-indicator" combinations may point to entirely different policy directions. For example, a pattern of high functional polycentricity identified based on ESDA and network density may support a "networked governance" policy aimed at strengthening regional coordination. Conversely, a strong monocentric structure identified based on GL_CS and primacy may point toward a "polycentric development" strategy focused on cultivating secondary growth poles. Accordingly, there is no universally applicable method, since different urban contexts may require distinct perspectives on polycentricity. This calls for deeper analytical reflection and nuanced understanding of the complexity of measurement and interpretation. Employing systematic and rigorous scientific methods is crucial for identifying spatial patterns accurately and avoiding erroneous conclusions. Misinterpreting the factors shaping polycentric patterns may lead to spurious findings and biases in subsequent research. In summary, this study emphasizes the necessity of continually re-examining and refining methodologies to enhance our understanding of the complex phenomenon of polycentricity. Any research on functional polycentricity must explicitly and thoroughly state the center identification methods, measurement indicators, and their theoretical foundations. When comparing results across studies, it is essential to first assess whether their methodological foundations are comparable.

It is important to note that our operationalization of functional polycentricity relies primarily on weekday commuting flows. While this captures a fundamental dimension of urban spatial interaction, it represents a subset of the city's total

functional rhythm. Urban life is equally defined by social, recreational, and service-oriented movements, which may exhibit different spatial patterns and center formations, particularly during evenings and weekends. Our analysis therefore captures the 'work-centric' functional structure, which is a critical but incomplete picture of polycentricity. Moreover, we acknowledge several inherent limitations of this research: restricted city sample selection, data availability, and indicator compatibility. The study covers only three cities, which limits the geographic scope, and the conclusions require further validation in megacities (e.g., Beijing, Shanghai) and small-to-medium-sized cities. The lack of temporal data prevents capturing the evolutionary process of polycentricity (e.g., the rise of Wuhan's Optics Valley). Currently, mobile signaling data analysis predominantly captures regular commuting patterns, while the reliable isolation and classification of fine-grained, low-frequency non-work trips—particularly those related to social and recreational activities—remains a methodological challenge. Consequently, the so-called "functional" centers identified in this study are more accurately described as "employment-commuting dominant centers." It is plausible that the hierarchical structure and interaction patterns among centers could differ substantially for non-work purposes. For instance, a sub-center that appears weak in terms of commuting flows might function as a prominent recreational or social hub. This limitation partly accounts for the observed divergence between morphological and functional polycentricity measures, as land-use patterns (morphology) often support mixed functions, whereas the flow data used here primarily reflect a single functional dimension. In future research, we explicitly propose extending the analysis to multi-purpose trip chains and weekend mobility data to achieve a more comprehensive understanding of urban functional structures. Some indicators, such as SRSD, rely on theoretical assumptions (e.g., spatial stationarity) that may conflict with the discontinuous, leapfrogging development observed in actual urban data. Future research on polycentricity should move beyond "morphology-only" or "method-driven" perspectives and instead adopt an integrated framework of "problem orientation – indicator alignment – method adaptation." Strengthening longitudinal analyses and cross-city comparisons will be essential for uncovering both the general laws and localized pathways of urban polycentric development.

## Acknowledgments

The authors would like to sincerely thank all colleagues and friends who voluntarily reviewed the translation of the survey and provided feedback on earlier drafts of this manuscript. We are especially grateful to Prof. Xinyi Niu for his invaluable insights and constructive suggestions during the revision of this paper.

## Author contributions

**Conceptualization:** Juan Zhu.

**Data curation:** Juan Zhu.

**Formal analysis:** Juan Zhu.

**Investigation:** Yao Wang.

**Methodology:** Juan Zhu.

**Writing – original draft:** Juan Zhu.

**Writing – review & editing:** Yao Wang.

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
