## [Decision Letter · Decision Letter 0]

9 Apr 2026

PONE-D-25-58646A Sensitivity Study on the Measurement of Urban Polycentricity in Chinese Cities: Center Definition, Indicator Selection, and Their Interaction EffectsPLOS One

Dear Dr. Zhu,

Thank you for submitting your manuscript to PLOS ONE. After careful consideration, we feel that it has merit but does not fully meet PLOS ONE’s publication criteria as it currently stands. Therefore, we invite you to submit a revised version of the manuscript that addresses the points raised during the review process.

We look forward to receiving your revised manuscript.

Kind regards,

Umberto Baresi, Ph.D.

Academic Editor

PLOS One

Journal Requirements:

The authors would like to acknowledge all colleagues and friends who have voluntarily reviewed the translation of the survey and the manuscript of this study. This research study was supported by general project of philosophy and social science research in universities of Jiangsu Province(2023SJYB1622).

The authors would like to acknowledge all colleagues and friends who have voluntarily reviewed the translation of the survey and the manuscript of this study. This research study was supported by general project of philosophy and social science research in universities of Jiangsu Province(2023SJYB1622).

The authors would like to acknowledge all colleagues and friends who have voluntarily reviewed the translation of the survey and the manuscript of this study. This research study was supported by general project of philosophy and social science research in universities of Jiangsu Province(2023SJYB1622).

7. Your abstract cannot contain citations. Please only include citations in the body text of the manuscript, and ensure that they remain in ascending numerical order on first mention.

8. Please ensure that you refer to Figure 1 in your text as, if accepted, production will need this reference to link the reader to the figure.

9. We note that Figure(s) 2, 3 in your submission contain [map/satellite] images which may be copyrighted. All PLOS content is published under the Creative Commons Attribution License (CC BY 4.0), which means that the manuscript, images, and Supporting Information files will be freely available online, and any third party is permitted to access, download, copy, distribute, and use these materials in any way, even commercially, with proper attribution. For these reasons, we cannot publish previously copyrighted maps or satellite images created using proprietary data, such as Google software (Google Maps, Street View, and Earth). For more information, see our copyright guidelines: http://journals.plos.org/plosone/s/licenses-and-copyright.

a. You may seek permission from the original copyright holder of Figure(s) 2, 3 to publish the content specifically under the CC BY 4.0 license.

10. We note that Figure 1 in your submission contain copyrighted images. All PLOS content is published under the Creative Commons Attribution License (CC BY 4.0), which means that the manuscript, images, and Supporting Information files will be freely available online, and any third party is permitted to access, download, copy, distribute, and use these materials in any way, even commercially, with proper attribution. For more information, see our copyright guidelines: http://journals.plos.org/plosone/s/licenses-and-copyright.

Reviewers' comments:

Reviewer's Responses to Questions

**Comments to the Author**

1. Is the manuscript technically sound, and do the data support the conclusions?

Reviewer #1: Yes

Reviewer #2: Yes

2. Has the statistical analysis been performed appropriately and rigorously? 

Reviewer #1: I Don't Know

Reviewer #2: Yes

3. Have the authors made all data underlying the findings in their manuscript fully available?

Reviewer #1: No

Reviewer #2: Yes

4. Is the manuscript presented in an intelligible fashion and written in standard English?

Reviewer #1: Yes

Reviewer #2: Yes

5. Review Comments to the Author

Reviewer #1: This is a robust study, addressing a topic that has been extensively explored and discussed through different theoretical and methodological approaches throughout decades. The article makes this clear and defines its methodological choices. However, it seems that a more in-depth discussion on the conceptualization of morphological and functional centrality would be appropriate, as to present the wide range of possibilities in literature. For instance, those connected to geotecnologies and spatial dynamic analysis - that seem to be better suitable for the dynamic reading required.

Although this study explores different methods and demonstrates some significant differences in its findings, its conclusion that there is a clear need for more dynamic ways of understanding urban centrality is well-founded. It offers interesting contributions to the field by comparatively examining specific methods and its performance/output.

The absence of certain data is justified due to their nature and the requirement for protection.

Reviewer #2: Advantages

1) Academic Originality and Filling the Research Gap

This paper makes a clear contribution by shifting focus from simply identifying polycentricity to critically examining how it is measured. The authors show that existing research is fragmented and lacks common standards, making cross-study comparison unreliable. This paper directly addresses that problem. The authors tackle this by using a Linear Mixed-effects Model (LMM) to quantify these interactions, moving the field from chasing a single definitive answer toward understanding where and why results diverge. Interestingly, this paper uses mobile signaling data to combine both static (morphological) and dynamic flow (functional) perspectives into one coherent framework, rather than analyzing only one dimension as most prior studies did.

2) Validity and Rigor of Characterizing Centers

The morphology-function distinction is a well-established framework. The authors apply it carefully and consistently throughout. Morphological Dimension is measured through the evenness of center sizes, using static data such as working population distribution, whereas Functional Dimension is measured through the relative importance of commuting flows, using dynamic mobile signal data to capture how people move between centers. The authors test their findings across 4 different center identification methods and 5 different indicators. This multi-method approach is designed to reveal how much results shift depending on the tools used. Additionally, the LMM isolates the effect of methodological choices from actual urban variation—a statistically sound approach for a multi-city comparative study.

Improvements

The authors have made great efforts to quantitative analysis. In part 5 and 6, interpretation and implications of the findings should be developed through these followings:

1. Strengthening the Functional Characterization

The reliance on commuting flows as the primary measure of functional polycentricity is a notable limitation. While commuting data is valuable, urban function extends well beyond employment-home linkages—social, recreational, and service-oriented movements are equally central to how cities actually operate. As it stands, the functional dimension captures only one slice of urban life. The authors should explicitly acknowledge this boundary, and where the data permits, incorporate non-work-related flows or at minimum discuss how the polycentric structure might shift during weekends or leisure periods. This would give a clearer picture of each city's functional rhythm.

2. Deepening the Socio-Economic Context of Case Studies

The 3 case cities—Hangzhou, Wuhan, and Nanning—are treated largely as interchangeable test subjects. Each city carries distinct geographical and policy histories that directly shape why certain measurement methods perform well or poorly within them. Hangzhou's West Lake constraint, Wuhan's river-split "Three Towns" structure, and Nanning's economic geography are not background details—they are explanatory factors. The authors should include a brief spatial narrative for each city, connecting local conditions to specific methodological outcomes. For instance, explaining why the threshold method overestimates centers in one city but underestimates them in another would bridge the gap between statistical results and planning reality.

3. Providing a Practical Guide for Practitioners

The paper demonstrates that different methods produce different results, but it stops short of telling readers what to do with that finding. A practitioner finishing this paper may be left more uncertain than before, unsure which method to trust for their specific context. The authors should translate their findings into a practical decision-making framework—such as a synthesis table—that guides method selection based on city type, urbanization stage, or measurement goal. This would shift the paper's contribution from diagnosing a problem to actively solving it, making it a genuinely useful reference for future researchers and planners.

4. Visualizing the Interaction Effects

The LMM results are statistically sound but risk being inaccessible to a broader planning audience when presented primarily through text and tables. Since the interaction between indicator selection and center definition methods is the core of the paper's originality, it deserves a visual treatment that makes that interaction immediately legible. A heatmap or conceptual matrix—showing which indicators are robust across all center definitions and which are highly sensitive to definitional choices—would make the key findings far more communicable and impactful.

5. Expanding the Policy Implications

The paper's implications for urban planning policy remain insufficiently developed. The stakes of methodological misclassification are not merely academic: if a city planner relies on a flawed method and incorrectly determines that a city is monocentric, the downstream consequences—misallocated infrastructure, poorly targeted investment zones, ineffective transport planning—can be significant. This is especially consequential in the Chinese context, where polycentric development is actively pursued as a top-down planning strategy. The authors should add a focused discussion on policy sensitivity, making explicit how methodological bias translates into planning risk.

6. PLOS authors have the option to publish the peer review history of their article (what does this mean?). If published, this will include your full peer review and any attached files.

Reviewer #1: **Yes:** Vânia Raquel Teles Loureiro

Reviewer #2: No

---

## [Author Response · Author response to Decision Letter 1]

7 May 2026

To the Editor and Reviewers:

We sincerely thank the Editor and the two Reviewers for their insightful and constructive comments on our manuscript. We have carefully considered all points raised and have revised the manuscript accordingly. Below, we provide a point-by-point response to each comment. Changes made to the manuscript are detailed, and the corresponding lines/paragraphs in the revised version are indicated for the Editor's and Reviewers' convenience. We believe the manuscript has been significantly improved and now fully meets PLOS ONE's publication standards.

Responses to the Editor’s Administrative Comments

Editor’s Comment 1: Please ensure that your manuscript meets PLOS ONE's style requirements, including those for file naming.

Response: We have meticulously formatted the manuscript according to the official PLOS ONE style templates. All submitted files have been renamed following the journal’s recommended naming conventions for clear identification and processing. This includes full compliance with guidelines for structure, font, spacing, margins, headings, and reference formatting.

Editor’s Comment 2: We note that you have indicated that there are restrictions to data sharing for this study. PLOS only allows data to be available upon request if there are legal or ethical restrictions on sharing data publicly.

Response: The core data used in this study derive from proprietary cellular signaling data provided by China Mobile Communications Group Co., Ltd. These data consist of anonymized, aggregated records of mobile device locations and movements at the traffic analysis zone level. Although processed to protect privacy, the dataset is considered commercially sensitive proprietary information. Our access and use are governed by a Data Licensing and Non-Disclosure Agreement with China Mobile, which expressly prohibits public deposition, redistribution, or open sharing of the raw or processed data. This is a standard contractual provision to safeguard the data owner’s business interests and competitive assets. Researchers seeking access to similar China Mobile signaling data for academic verification should contact the data owner directly.

Editor’s Comment 3: Please provide a complete Data Availability Statement in the submission form, ensuring you include all necessary access information or a reason for why you are unable to make your data freely accessible.

Response:

Data Availability: The dataset underlying this study consists of proprietary cellular signaling data provided by China Mobile Communications Group Co., Ltd. This dataset contains detailed, aggregated records of mobile device locations and movements. Public sharing of this dataset is restricted due to a legally-binding Data Licensing and Non-Disclosure Agreement between the data provider and our research institution. The agreement explicitly prohibits the redistribution or public deposition of the raw or processed signaling data to protect commercial confidentiality and user privacy. Researchers interested in the underlying data may contact China Mobile directly (https://www.10086.cn).

Editor’s Comment 4: When completing the data availability statement of the submission form, you indicated that you will make your data available on acceptance. We strongly recommend all authors decide on a data sharing plan before acceptance, as the process can be lengthy and hold up publication timelines. Please note that, though access restrictions are acceptable now, your entire data will need to be made freely accessible if your manuscript is accepted for publication. This policy applies to all data except where public deposition would breach compliance with the protocol approved by your research ethics board. If you are unable to adhere to our open data policy, please kindly revise your statement to explain your reasoning and we will seek the editor's input on an exemption. Please be assured that, once you have provided your new statement, the assessment of your exemption will not hold up the peer review process.

Response: Thank you for your clear and important guidance regarding PLOS ONE’s open data policy and the pre-acceptance data sharing plan. We have carefully reviewed the policy and the specific contractual terms governing our primary dataset. We write to formally request an exemption from the requirement to make the entire dataset freely accessible upon publication, as the public deposition of our core data would constitute a breach of a legally-binding third-party agreement, which aligns with the exception noted in your policy (“where public deposition would breach compliance with the protocol approved by your research ethics board”).The data are derived from China Mobile under a commercial Data Licensing and Non-Disclosure Agreement that explicitly prohibits public deposition or redistribution. Public sharing would constitute a breach of this binding contract, aligning with the policy exception for cases where deposition would violate an approved protocol. In this instance, the “protocol” is the legally-binding agreement with the data owner. We appreciate your understanding of this complex situation and await the editor’s decision regarding this exemption request. The peer review of our manuscript’s scientific content can proceed unimpeded.

Editor’s Comment 5: Thank you for stating the following financial disclosure: The authors would like to acknowledge all colleagues and friends who have voluntarily reviewed the translation of the survey and the manuscript of this study. This research study was supported by general project of philosophy and social science research in universities of Jiangsu Province(2023SJYB1622). Please state what role the funders took in the study. If the funders had no role, please state: "The funders had no role in study design, data collection and analysis, decision to publish, or preparation of the manuscript." If this statement is not correct you must amend it as needed. Please include this amended Role of Funder statement in your cover letter; we will change the online submission form on your behalf.

Response: Please update the Funding Statement in the online submission form to: “This work was supported by the General Project of Philosophy and Social Science Research in Universities of Jiangsu Province (Grant No. 2023SJYB1622). The funders had no role in study design, data collection and analysis, decision to publish, or preparation of the manuscript.”

Editor’s Comment 6: Thank you for stating the following in the Acknowledgments Section of your manuscript: The authors would like to acknowledge all colleagues and friends who have voluntarily reviewed the translation of the survey and the manuscript of this study. We note that you have provided funding information that is not currently declared in your Funding Statement. However, funding information should not appear in the Acknowledgments section or other areas of your manuscript.

Response: We have removed all funding-related text from the Acknowledgments section. The revised section now reads: “The authors sincerely thank all colleagues and friends who voluntarily reviewed the survey translation and provided feedback on earlier drafts. We are especially grateful to Prof. Xinyi Niu for his invaluable insights and constructive suggestions during the revision of this paper.” The updated funding statement is provided in response to Comment 5.

Editor’s Comment 7: Your abstract cannot contain citations. Please only include citations in the body text of the manuscript, and ensure that they remain in ascending numerical order on first mention.

Response: We have confirmed that the abstract contains no citations. All references are placed within the main text and appear in ascending numerical order upon first mention. The abstract remains unchanged as it was already compliant.

Editor’s Comment 8: Please ensure that you refer to Figure 1 in your text as, if accepted, production will need this reference to link the reader to the figure.

Response: We have added a reference to Figure 1 in the main text. The revised sentence in the Introduction (line 99-100) reads: “See Figure 1,” ensuring proper linking for production.

Editor’s Comment 9: Concerns regarding potential copyright of map/satellite images in Figures 2, and 3, and the requirement for written permission or replacement to comply with the CC BY 4.0 license.

Response: We confirm that Figures 2 and 3 do not contain copyrighted imagery from proprietary sources such as Google Maps/Earth. The basemaps are derived from OpenStreetMap data (licensed under ODbL, compatible with CC BY 4.0). Administrative and TAZ boundaries are sourced from official municipal databases and used with permission. The figure captions have been updated to include attribution: “Basemap data from OpenStreetMap (© OpenStreetMap contributors, ODbL). Administrative and TAZ boundaries sourced from official municipal databases of Hangzhou, Wuhan, and Nanning and used with permission.” These are original schematic diagrams created by the authors.

Editor’s Comment 10: We note that Figure 1 in your submission contain copyrighted images. All PLOS content is published under the Creative Commons Attribution License (CC BY 4.0), which means that the manuscript, images, and Supporting Information files will be freely available online, and any third party is permitted to access, download, copy, distribute, and use these materials in any way, even commercially, with proper attribution.

Response: Figure 1 is an original schematic diagram created by the authors using Adobe Illustrator, consisting solely of author-generated elements. It contains no third-party copyrighted images, maps, or satellite imagery. As an original work, it is fully compatible with the CC BY 4.0 license and presents no copyright concerns.

Editor’s Comment 11: If the reviewer comments include a recommendation to cite specific previously published works, please review and evaluate these publications to determine whether they are relevant and should be cited. There is no requirement to cite these works unless the editor has indicated otherwise.

Response: We have reviewed all reviewer comments and confirm that neither reviewer recommended citing any specific previously published works. Therefore, no new citations have been added based on reviewer suggestions.

Editor’s Comment 12: Please review your reference list to ensure that it is complete and correct.

Response: We have carefully reviewed and updated the entire reference list in accordance with the revisions made to the manuscript. We confirm that the list is now complete and accurate. All in-text citations have been verified to correspond correctly with the entries in the reference list, and the formatting is consistent with the journal's style guide. No retracted articles are cited.

Responses to Reviewers’ Comments

Responses to Reviewer #1:

Comment: The reviewer acknowledges the robustness and contribution of the study but suggests a deeper discussion on the conceptualization of morphological and functional centrality, particularly referencing geotechnologies and spatial dynamic analysis.

Response: We sincerely thank the reviewer for the positive assessment of our work and for the constructive suggestion to deepen the conceptual discussion. We agree that providing a stronger theoretical foundation, especially by linking to modern geotechnologies and dynamic analysis methods, will significantly strengthen the paper's framing and contribution. In direct response to this valuable feedback, we have significantly expanded the theoretical discussion in the Literature Review sections 2.2 of the revised manuscript (around lines 198-237 in the revised manuscript).

Responses to Reviewer #2:

Comment 1: The reviewer points out that relying solely on commuting (work-home) flows to measure functional polycentricity is a limitation, as urban functions encompass a wider range of movements (social, recreational, service-oriented). They suggest acknowledging this boundary and, if possible, incorporating non-work flows or discussing potential variations during non-work periods (e.g., weekends).

Response: We thank the reviewer for this crucial and insightful observation. We completely agree that functional polycentricity, in its most comprehensive sense, should reflect the full spectrum of urban activities. The reviewer’s comment rightly identifies a key conceptual boundary in our current operationalization. In the part 6 of Conclusion section (see lines 916-922,927-940), we have expanded our interpretation of the results to consider the implications of this limitation.

Comment 2: The reviewer astutely observes that the three case cities were initially presented as somewhat interchangeable test subjects, and suggests enriching the analysis by providing a brief spatial narrative for each that connects their unique geographical, historical, and policy contexts to the specific methodological outcomes observed in the study.

Response: We sincerely thank the reviewer for this excellent suggestion. We agree that the distinct urban DNA of each city is not merely background but a crucial lens through which to interpret our quantitative findings. Incorporating this context significantly strengthens the paper’s explanatory power and practical relevance.

Action Taken: In direct response, we have substantially revised the Results and analysis section 4.1.3 (now located in the manuscript around lines 493-561).

Comment 3:The reviewer insightfully notes that while the paper demonstrates methodological variability, it should go further to provide clear, actionable guidance for practitioners on how to choosea method. They suggest developing a practical decision-making framework (e.g., a synthesis table) to help users select methods based on city type, urbanization stage, or research goal, thereby shifting the contribution from problem diagnosis to problem-solving.

Response: We sincerely thank the reviewer for this excellent and constructive suggestion. We completely agree that translating our complex methodological findings into clear, practical guidance is crucial for maximizing the paper’s impact and utility for researchers and planners. In direct response, we revised Discussion 5.1/5.2 section (see lines 733-761/ 782-795 in the revised manuscript). The core of this addition is a new synthesis table (Table 8), which serves as the practical decision-making framework suggested by the reviewer.

Comment 4: The reviewer astutely notes that the statistically robust LMM results, when presented only in text and tables, may be less accessible to a broader audience, including planners. They suggest that the core finding of “method-indicator” interaction deserves a visual treatment, such as a heatmap or conceptual matrix, to make the patterns immediately clear and impactful.

Response: After carefully considering options for visualizing the complex "method-indicator" interaction effects from the Linear Mixed Model (LMM), we found that generating a single, comprehensive heatmap presented significant challenges. This is primarily because our result data structure is multidimensional: it simultaneously involves four center identification methods, five measurement metrics, three case cities, and both morphological and functional dimensions. Compressing all this information into one heatmap could lead to information overload, potentially obscuring the core patterns rather than highlighting them. We highly value and have fully adopted the reviewer’s core suggestion to "enhance visual presentation." In the revised manuscript, we have modified the original box plot (Figure 4) to enable readers to quickly grasp the key patterns of the interaction effects.

Comment 5: The reviewer rightly points out that the policy implications of our findings, particularly regarding the risks of methodological misclassification in planning practice, require deeper development. They emphasize that the consequences of using a flawed method to assess a city’s structure are not just academic but can lead to significant real-world planning failures, especially in the Chinese context where polycentric development is a major strategic goal.

Response: We sincerely thank the re

---

## [Editor Report · Decision Letter 1]

11 May 2026

A sensitivity study on the measurement of urban polycentricity in Chinese cities: center definition, indicator selection, and their interaction effects

PONE-D-25-58646R1

Dear Dr. Zhu,

We’re pleased to inform you that your manuscript has been judged scientifically suitable for publication and will be formally accepted for publication once it meets all outstanding technical requirements.

Kind regards,

Umberto Baresi, Ph.D.

Academic Editor

PLOS One
---

## [Editor Report · Acceptance letter]

PONE-D-25-58646R1

PLOS One

Dear Dr. Zhu,

I'm pleased to inform you that your manuscript has been deemed suitable for publication in PLOS One. Congratulations! Your manuscript is now being handed over to our production team.

Kind regards,

on behalf of

Dr. Umberto Baresi

Academic Editor

PLOS One